# Exploring negative emission potential of biochar to achieve carbon neutrality goal in China

Xu Deng [1], Fei Teng [1] ✉, Minpeng Chen [2], Zhangliu Du[3], Bin Wang[4], Renqiang Li[5] & Pan Wang[5]

Limiting global warming to within 1.5 °C might require large-scale deployment of premature negative emission technologies with potentially adverse effects on the key sustainable development goals. Biochar has been proposed as an established technology for carbon sequestration with co-benefits in terms of soil quality and crop yield. However, the considerable uncertainties that exist in the potential, cost, and deployment strategies of biochar systems at national level prevent its deployment in China. Here, we conduct a spatially explicit analysis to investigate the negative emission potential, economics, and priority deployment sites of biochar derived from multiple feedstocks in China. Results show that biochar has negative emission potential of up to 0.92 billion tons of $CO_2$ per year with an average net cost of US\$90 per ton of $CO_2$ in a sustainable manner, which could satisfy the negative emission demands in most mitigation scenarios compatible with China's target of carbon neutrality by 2060.

Anthropogenic forcing caused warming of 0.9–1.3 °C during 2010–2019 relative to the preindustrial period[1], revealing the scale of the growing challenge in meeting the 1.5 or 2 °C warming climate goals specified in the Paris Agreement[2,3]. To achieve the stated climate goals, mitigation strategies increasingly rely on negative emission technologies (NETs) that can remove $CO_2$ from the atmosphere[4]. For example, for China to reach carbon neutrality, NETs are expected to provide negative emissions of 0.01–2.91 billion tons of $CO_2$ per year ($GtCO_2$ $yr^{-1}$) between 2050 and 2060, according to climate mitigation scenarios based on leading integrated assessment models (IAMs)[5–8]. In addition to the carbon sinks of reforestation and afforestation, these scenarios suggest that such high NET demands rely mostly on bioenergy with carbon capture and storage (BECCS) or direct air carbon capture and storage (DACCS)[9]. However, both BECCS and DACCS have financial and technological obstacles that must be overcome before they can be implemented on a broad scale[10,11]. Specifically, NET costs

(levelized cost per ton of $CO_2$ removed) can exceed US\$100 per ton of $CO_2$ (\$ $t^{-1}CO_2$) for BECCS and 200 \$ $t^{-1}CO_2$ for DACCS[12–14], while resource and geological constraints further limit their deployment[15]. Moreover, BECCS competes with crop production for both land and water, thereby potentially compromising other sustainable development goals such as food security[16]. Therefore, prior to addressing the major barriers confronting mentioned NETs, alternative solutions are urgently needed to form a feasible NET portfolio capable of achieving climate targets safely and sustainably[17].

Biochar represents a technically proven solution for realizing negative emissions, together with co-benefits in terms of soil fertility and crop productivity[18]. Biochar has ancient origins in Amazonian farmlands and it has existed for centuries[19,20], although it has gained recognition in the field of climate change mitigation in only the previous two decades[21,22]. At temperatures between 350 and 900 °C, slow pyrolysis converts biomass into less-degradable biochar, together with

[1]Institute of Energy, Environment and Economy, Tsinghua University, Beijing 100084, China. [2]School of Agricultural Economics and Rural Development, Renmin University of China, Beijing 100872, China. [3]College of Resources and Environmental Sciences, China Agricultural University, Beijing 100193, China. [4]Institute of Environment and Sustainable Development in Agriculture, Chinese Academy of Agricultural Sciences, Beijing 100081, China. [5]Key Laboratory of Ecosystem Network Observation and Modelling, Institute of Geographic Sciences and Natural Resources Research, Chinese Academy of Sciences, Beijing 100101, China. ✉e-mail: tengfei@tsinghua.edu.cn

by-products such as syngas[23]. Biochar can then be returned to the soil, which helps sequester carbon, avoid emission of soil greenhouse gases, and improve soil quality and crop yields[24]. China accords particular importance to practical use of biochar and a number of pilot projects have been conducted since the early 2010s[25]. In recent years, more than 100 companies in China have participated in the biochar business, with demonstration sites selected across the major crop-production areas[26]. However, biochar has largely been viewed as only a green agricultural technology that can reduce fertilizer input, build soil quality, and increase crop yields, and its role in terms of climate mitigation has largely been ignored.

Global evaluations of biochar's potential have underscored its critical function as a negative emission technology[18,27,28]. However, to fully harness its benefits, it is crucial to evaluate the negative emissions potential and economic viability of biochar at regional levels. A comprehensive spatial analysis integrating the latest knowledge on biochar's role in negative emissions is in need to provide actionable insights for its deployment in the pursuit of carbon neutrality. On one hand, existing experiments tends to narrowly focus on the properties of specific biochar types or their comparative analysis[29,30] without a granular estimation of their potential for negative emissions and economic impact. On the other hand, regional studies often fail to account for the diversity of biomass resources and biochar properties, leading to a flawed foundation for deployment strategies.

In China, estimates on biochar have predominantly focused on agricultural residues[31–33], neglecting significant contributions from forestry and grass residues, as well as potential energy crops. This oversight results in a chronic underestimation of the country's total biochar potential. Furthermore, variations in the physicochemical properties of biochar derived from different biomass resources[34,35] are typically overlooked, leading to inaccuracies in economic and emissions assessments. The heterogeneity of spatial factors, such as soil texture, is also commonly disregarded, resulting in biased crop yield benefit estimates[36,37] and flawed economic evaluations. Recent advancements in data availability from field experiments[38], assessment methodologies[39], and spatial data resolution[40] now permit the inclusion of biomass and spatial heterogeneity in assessments, facilitating a detailed and location-specific evaluation of biochar's negative emissions potential and economic implications. Our study leverages the latest scientific progress to present a spatially explicit analysis of biochar potential in China. This analysis recognizes the diversity in

biomass types and spatial distribution, addressing the prevalent underestimation of biochar's potential and providing a detailed, actionable framework for policymakers to guide biochar deployment.

In this study, we investigate the negative emission potential of biochar produced from multiple feedstocks and identify the most cost-effective biomass types and deployment locations in China. First, we evaluate the magnitude of available biomass feedstocks for biochar production, including biomass residues from agriculture, forest, grassland, and potential energy crops in the marginal land (as shown in the "Methods" section), and develop three scenarios based on various assumptions of biomass availability. Then, we quantify the negative emission potential of biochar using a uniform empirical framework, which takes into account biochar properties and pyrolysis parameters. Second, incorporating both literature-based and practical survey data, we conduct a cost-benefit analysis and construct supply curves of the negative emissions for biochar derived from multiple feedstocks. Finally, we make the spatially explicit analysis of the negative emission potential and economics to prioritize biochar deployment. Results show that biochar can achieve negative emission potential of up to 0.92 Gt $CO_2$ $yr^{-1}$ with an average cost of approximately 90 \$ $t^{-1}CO_2$ in a sustainable manner. Such potential of biochar could satisfy the negative emission demands in most mitigation scenarios compatible with China's target of carbon neutrality by 2060. Furthermore, we discover that feedstocks and subregions with high negative emission potential and high economics largely overlap, which could provide guidance for systematic deployment of biochar in China.

## Results
### Negative emission potential of biochar
We construct three scenarios to estimate the potential for biochar to act as a negative emission technology based on various assumptions of biomass availability (refer to the "Methods" section and Table 1). The first scenario, designated as the 'Maximum Theoretical Potential', entails the exploitation of marginal lands for energy crop cultivation and full utilization of available biomass for biochar production, serving as a benchmark compared with other studies investigating biomass potential in China. The 'Current Technical Potential' scenario, in contrast, limits biomass access to feedstocks harvestable through current technologies and practices without competing with current usages such as livestock feed and rural energy consumption. Lastly, the 'Sustainable Technical Potential' scenario foresees the cultivation of

**Table 1 | Potentials of available biomass feedstocks under various scenarios**

| Scenarios | Maximum theoretical potential | Sustainable technical potential | Current technical potential |
|---|---|---|---|
| Description | Maximum amount of available biomass feedstocks | Theoretical potential minus use for livestock, etc., while maintaining ecology | Available biomass feedstocks limited by current technology and practice |
| Agricultural residues | 0.99 Gt $yr^{-1}$ | 0.79 Gt $yr^{-1}$ | 0.73 Gt $yr^{-1}$ |
| | 100% of 16 types of crop residues | 95% of crop residues those are not used as feed, substrate and raw material | 88% of residues those are not used as feed, substrate and raw material |
| Forestry residues | 0.29 Gt $yr^{-1}$ | 0.23 Gt $yr^{-1}$ | 0.08 Gt $yr^{-1}$ |
| | 100% of 10 types of forestry residues | 80% of 10 types of forestry residues | 28% of 10 types of forestry residues |
| Grass residues | 0.29 G $yr^{-1}$ | 0.04 Gt $yr^{-1}$ | 0.00 Gt $yr^{-1}$ |
| | 100% of hay production in natural grasslands | Hay production in natural grasslands not used as feed | Not available under current technology |
| Energy crops | 0.86 Gt $yr^{-1}$ | 0.66 Gt $yr^{-1}$ | 0.00 Gt $yr^{-1}$ |
| | Maximum production potential of dedicated energy crops; Marginal land refers to shrub land, the intertidal zone, bottomland, and unused land | Limited production potential of dedicated energy crops; Marginal land refers to unused land and shrub land | Not available under current technology |
| Total biomass feedstocks | 2.43 Gt $yr^{-1}$ | 1.73 Gt $yr^{-1}$ | 0.81 Gt $yr^{-1}$ |

The scenarios represent the maximum theoretical potential, sustainable technical potential, and current technical potential, respectively; the biomass feedstocks include agricultural residues, forestry residues, grass residues, and potential dedicated energy crops, respectively.

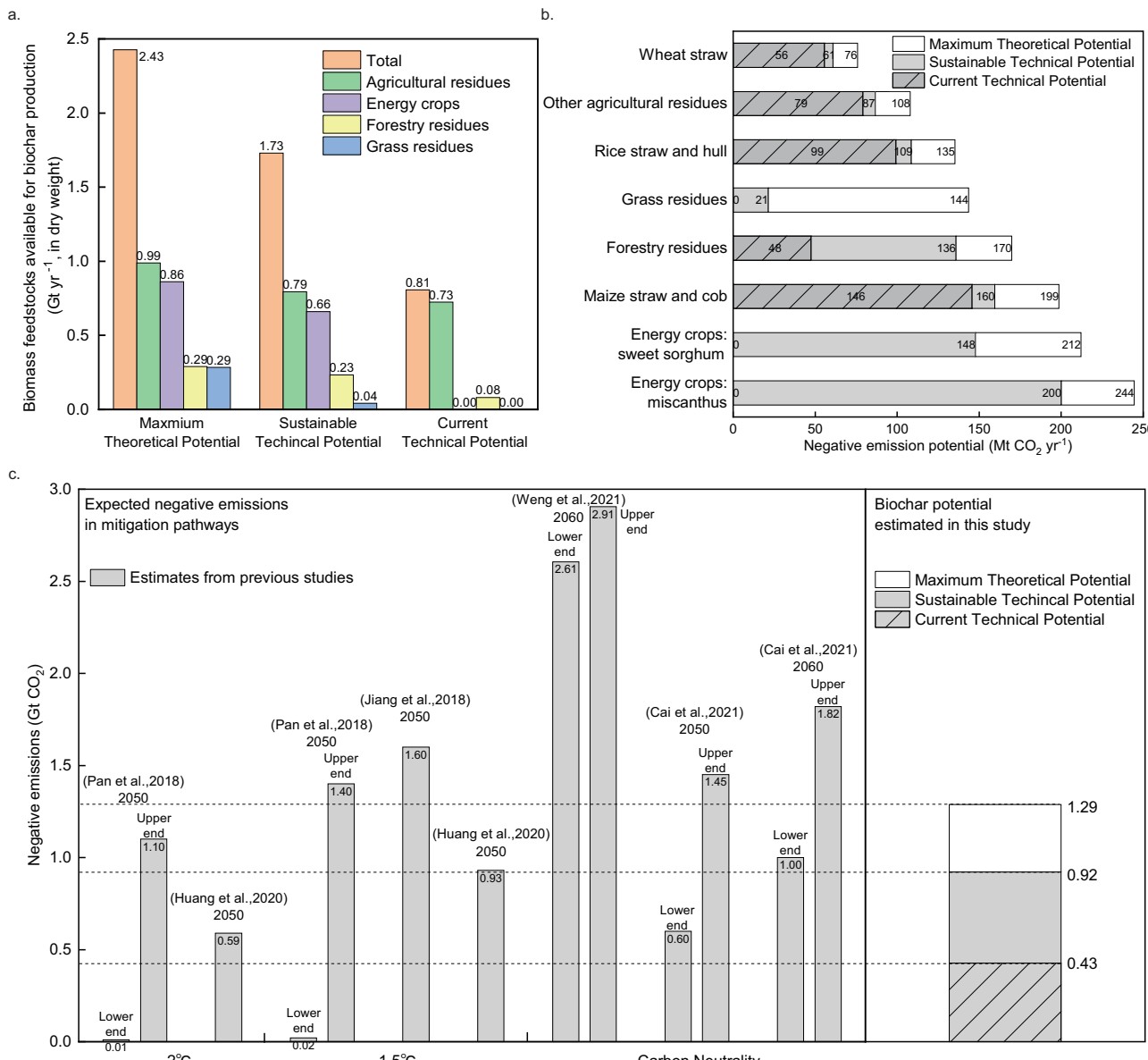

**Fig. 1 | Available biomass feedstocks and negative emission potential of biochar in China. a** Potentials of available biomass feedstocks under various scenarios, namely, maximum theoretical potential, sustainable technical potential, and current technical potential. These feedstocks include agricultural residues, forestry residues, grass residues, and potential dedicated energy crops. **b** Negative emission potential of biochar produced from various types of feedstocks. Note that 'other agricultural residues' refers to the residues of 13 other types of crops excluding rice, wheat, and maize. **c** Comparison between negative emission demands in mitigation pathways from previous studies, and the negative emission potential of biochar estimated in this study. Source data are provided as a Source data file.

energy crops on marginal lands, a more efficient residue harvesting rate brought about by technological and policy progress, while maintaining current biomass use patterns. Therefore, the 'Current Technical' and 'Sustainable Technical' scenarios present near-term and sustainable upper limits of biochar's negative emission potential without compromising food security or habitats. However, it's crucial to note that both scenarios necessitate progressive policy action to enhance biomass availability beyond current practices. The range of negative emission potentials reported in this paper pertains solely to the scenarios considered and does not reflect constraints imposed by real-world policies.

Our results indicate that the maximum theoretical potential of biomass feedstocks in China could reach as high as 2.43 Gt yr⁻¹ (Fig. 1a). Agricultural residues, predominantly composed of maize straw and cob, rice straw and hull, and wheat straw, contribute 41% or 0.99 Gt yr⁻¹ of the total feedstocks (Supplementary Fig. 9). In addition, 36% of the

total feedstocks, equivalent to 0.86 Gt yr⁻¹, originate from dedicated energy crops such as miscanthus and sweet sorghum grown in marginal lands. Forestry and grassland contribute equally to the total residues, each offering 0.29 Gt yr⁻¹. Compared to existing studies[40,41], our estimation on the maximum theoretical potential of available feedstocks is conservative, because we consider only residues in agriculture and forest biomass, grass in grassland excluding nature reserves, and potential energy crops limited by marginal lands and climatic conditions. Nonetheless, the abundance of biomass feedstocks provides great confidence for biochar preparation.

Considering the competition for biomass from current uses, such as livestock feed, along with ecological constraints, the total available biomass feedstocks under the 'Sustainable Technical Potential' scenario are diminished to 1.73 Gt yr⁻¹. Specifically, upon factoring in the harvesting rate and subtracting feedstocks utilized for livestock, rural energy consumption, and raw materials, the residue amounts available

from agriculture and forestry for biochar production are calculated at 0.79 Gt yr$^{-1}$ and 0.23 Gt yr$^{-1}$, respectively (refer to Supplementary Note 1). Grass residues, extensively used as livestock feed (see the "Methods" section), exhibit a notable reduction in availability. Our results demonstrate that the remaining grass residues have declined to 43 Mt yr$^{-1}$, representing only 15% of the maximum theoretical potential. Furthermore, to avoid ecological harm, this study assumes that dedicated energy crop cultivation is prohibited in intertidal zones, bottomlands, and certain government-designated shrublands. Consequently, the ensuing area for energy crop cultivation is -50.5 Mhm$^2$, yielding 0.66 Gt yr$^{-1}$. These figures align with existing estimates, which range from 3–185 Mhm$^2$ [42–44] for area and 0.01 to over 1 Gt yr$^{-1}$ [44–47] for production. Finally, under the 'Current Technical Potential' scenario, the available biomass feedstocks amount to 0.81 Gt yr$^{-1}$, comprised of 0.73 Gt yr$^{-1}$ from agricultural biomass and 0.08 Gt yr$^{-1}$ from forestry residues.

After pyrolysis, a portion of carbon from the biomass will be sequestered within the biochar and preserved for hundreds of years (Supplementary Note 3). The potential for negative emissions could reach 1.29 GtCO$_2$ yr$^{-1}$ under the maximum theoretical scenario. This capacity nearly fulfils the negative emission requirements across all deep decarbonization pathways in line with the 2 °C target (see Fig. 1b, c). Under the current technical scenario, the negative emission potential amounts to 0.43 GtCO$_2$ yr$^{-1}$, thereby presenting significant near-term mitigation opportunities. The potential under sustainable technical scenario, meanwhile, could fulfill a negative emission demand of 0.92 GtCO$_2$ yr$^{-1}$. Given that the median projection for negative emission demands in China is 1.43 GtCO$_2$ yr$^{-1}$ in 2050 or 2060 (see Fig. 1c), combined with the carbon sink in managed forests being 0.63Gt CO$_2$ yr$^{-1}$ [48], biochar stands to play a significant role in achieving negative emissions in accordance with the 1.5 °C target and carbon neutrality, without deploying premature NETs such as BECCS and DACCS.

Beyond providing negative emissions through carbon sequestration, biochar application has greater impact on the reduction in total emissions, e.g., by offsetting fossil carbon emissions through use of syngas, as well as avoiding soil greenhouse gas emissions (Supplementary Note 6). Our study suggests that the total avoidance part is approximately 1.5 times greater than the removal part (Supplementary Fig. 12), which indicates that biochar could play a greater role in climate mitigation. We also conducted the uncertainty analysis to show the long-term impact of climate change on the estimation of biochar potential. Without additional consideration for land use pattern, the Monte Carlo analysis suggests a slight growth of the negative emission potential of biochar under all scenarios. This growth is primarily attributable to the rise in forestry residues, which are affected by climate change expected in the second half of the 21st century[49–51] (Supplementary Table 13). Socio-economic factors also influence future crop production and, consequently, the availability of crop residues, showing significant variations (Supplementary Note 7).

### Economics of negative emissions from biochar
Then, we conduct the cost-benefit analysis on the slow pyrolysis (as shown in the "Methods" section) and construct the supply curves for costs and net costs (with by-products and yields increasing co-benefits as income) of negative emissions provided by biochar (Fig. 2a, b). The average net negative emission cost of biochar in China is 90 \$ t$^{-1}$CO$_2$, ranging from 60-96 \$ t$^{-1}$CO$_2$ for biochar derived from agricultural and forestry residues to 101−144 \$ t$^{-1}$CO$_2$ for biochar derived from energy crops and grass residues. Accordingly, the net cost of negative emissions for biochar from agricultural and forestry residues is capped at <100 \$ t$^{-1}$CO$_2$ in China, while the net cost for BECCS is typically 30−400 \$ t$^{-1}$CO$_2$ [12]. Although biochar from energy crops and grass residues is more expensive due to high biomass purchasing cost, they still has an economic advantage over other NETs, such as CO$_2$ fuels (0−670 \$

t$^{-1}$CO$_2$), DACCS (30−1000 \$ t$^{-1}$CO$_2$), and microalgae (230−920 \$ t$^{-1}$CO$_2$) that might be even more costly[12,13]. Biochar production technology, particularly the technology of biomass gasification for biochar and syngas co-production that is promoted in China (Supplementary Note 2), is both commercially mature and economically competitive, as evidenced by our results, suggesting that biochar could be regarded as a relatively cost-effective NET and that it has potential to play a key role in climate mitigation strategies.

The relatively low net cost of biochar is achieved owing to the sale of by-products syngas and the co-benefits from improved yields (left half of Fig. 2c). Syngas sales dominate the total revenue and offsets 32% to 57% of total costs. The benefits derived from by-product sales are least for forestry and rice residues. The carbon content of forestry-residue-based biochar is high, indicating that more carbon and heat are sequestered in the biochar rather than in the by-products. The heating value of rice straw and husk is low, indicating less syngas production when compared to other feedstocks (Supplementary Note 2). In current pilot projects, syngas is used for industrial heat or electricity generation in facilities near the pilot sites (Supplementary Table 9). Therefore, to scale-up the revenues from syngas sales, the expansion and enhancement of related infrastructure, such as improved gas and power grid connectivity, are crucial. Biochar also offers additional application incentives through its ability to bolster crop yields. However, the estimated co-benefits from yield improvements in our study are minor, offsetting only 1% to 23% of costs. This may be due to the prevalent high-rate fertilizer application in Chinese fields, which makes the yield enhancement impact of biochar comparatively less significant than suggested by international studies. For instance, the latest meta-analysis indicates that biochar application only improves major crop yields in China by approximately 10%[31], a figure significantly lower than the global average of 35%[36]. Conversely, the yield improvement effect is more noticeable for herbaceous-based biochars[36], such as those derived from miscanthus.

Without considering the revenue from yield improvements and by-product sales, the economic attractiveness of biochar would be greatly reduced, with negative emission costs rising to 142−273 \$ t$^{-1}$CO$_2$ (Fig. 2b). The negative emission cost of biochar derived from different feedstocks varies widely owing to differences in the cost of purchasing feedstock, the conversion rate from feedstock to biochar, and the carbon content in the biochar. First, biochar derived from energy crops is more expensive because the purchasing price of energy crops is higher than that of crop residues, since the latter does not include revenue from crop production. Second, a lower conversion rate from biomass to biochar or a lower carbon content in the biochar contributes to a higher total cost per unit of negative emission. For example, the conversion rates from energy crops to biochar are <25% (Supplementary Table 6), resulting in their high negative emission costs of over 200 \$ t$^{-1}$CO$_2$. In contrast, the conversion rates of rice, maize, and wheat residues to biochar are high under the same pyrolysis conditions, and their negative emission costs are 158, 162, and 168 \$ t$^{-1}$CO$_2$, respectively. Owing to the high carbon content of forest-derived biochar (77.2%), the negative emission cost of forestry residues is 142 \$ t$^{-1}$CO$_2$, which remains the lowest in all biochar types. Given the high upfront input and uncertain returns, biochar applications could begin with the collection of agricultural and forestry residues characterized by high carbon content and conversion rates.

### Spatial analysis of biochar potential
To identify areas suitable for biochar deployment, spatial analysis of negative emission potential under 'Sustainable Technical Potential' scenario is performed (see the "Methods" section) and the provinces are divided into six regions, as shown in Fig. 3. Because the feedstocks are not distributed uniformly throughout the regions, substantially diverse distribution patterns for negative emission potential are presented. Agricultural residues and energy crops dominate the

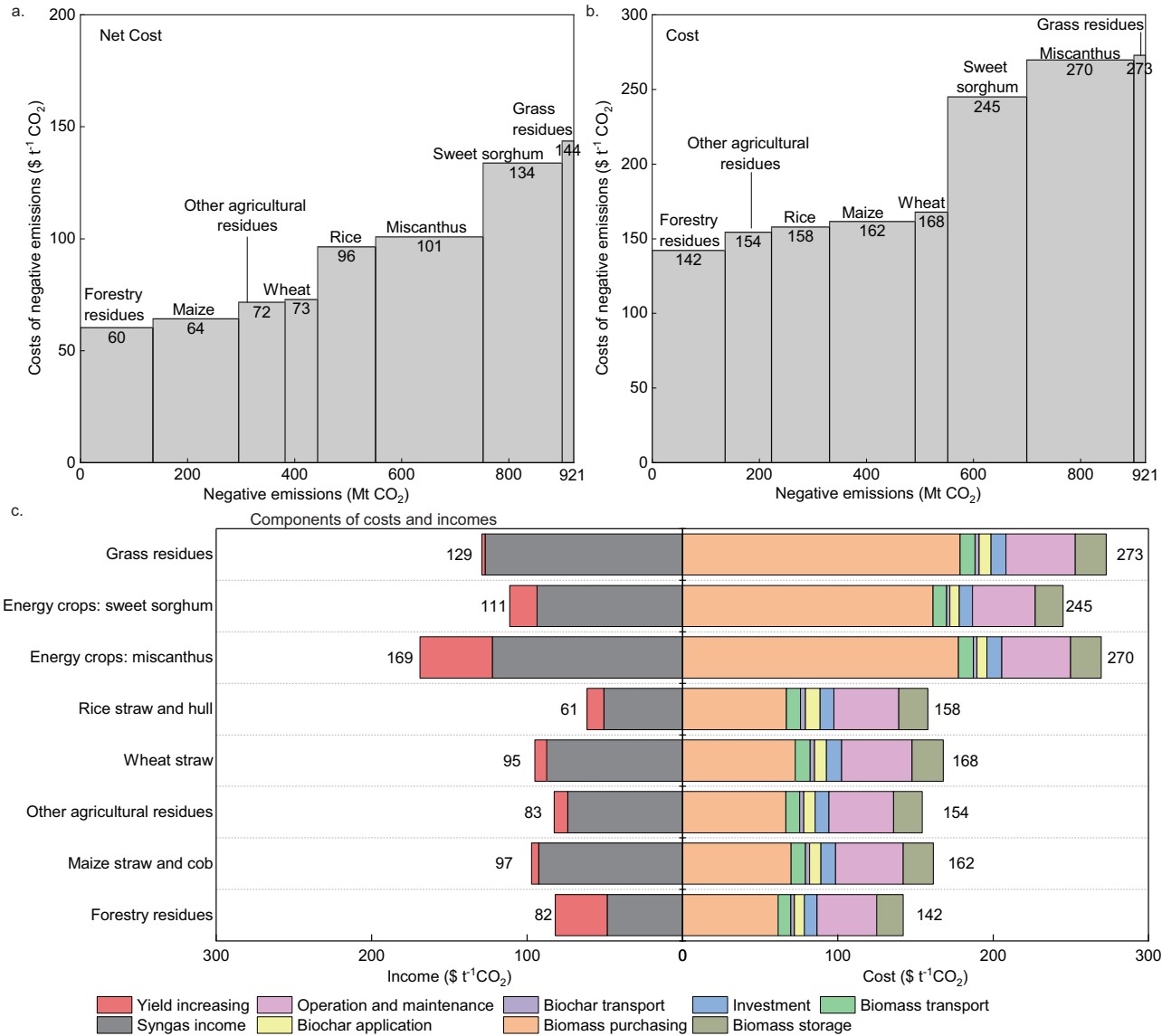

**Fig. 2 | Economics of negative emissions from biochar. a** The cost curve ($ $t^{-1}CO_2$) of the negative emissions of biochar derived from various feedstocks. Costs consist of the cost of feedstock purchasing, feedstock storage, investment, operation and maintenance, transport of biomass and biochar, and biochar application. **b** The net cost curve ($ $t^{-1}CO_2$) of the negative emissions of biochar derived from various feedstocks. Net cost refers to the difference between costs and incomes. Incomes consist of increased yield income and syngas income. **c** Components of costs and incomes for various types of biochar. Source data are provided as a Source data file.

distribution patterns because they are the most abundant biomass feedstocks. Agricultural residues are primarily distributed in Central and South China (191 Mt $yr^{-1}$), whereas energy crops are primarily distributed in North China (167 Mt $yr^{-1}$) and Southwest China (218 Mt $yr^{-1}$). Northwest China has a significantly lower negative emission potential than other regions because of its large proportion of grassland and absence of agricultural residues. Therefore, Central and South China has the greatest negative emission potential of 207 $MtCO_2$ $yr^{-1}$, followed by Southwest China (194 $MtCO_2$ $yr^{-1}$) and North China (161 $MtCO_2$ $yr^{-1}$) (Fig. 3j). Northwest China has the lowest negative emission potential, which is 87 $MtCO_2$ $yr^{-1}$.

Negative emission costs of biochar vary greatly at the regional level from 12–150 $ $t^{-1}CO_2$ (Fig. 4a) resulting from different feedstock types, biomass abundance, and soil types and PH (see "Methods" section). First, biochar derived from forestry and agricultural residues is the cheapest option for carbon removal. Thus, the cost of biochar is lower in regions rich in agroforestry residues (Fig. 4b, c). For example, the share of agroforestry residue in total available biomass feedstocks is 85% in East China and 70% in Central and South China, with low average costs of 77.8 and 78.3 $ $t^{-1}CO_2$, respectively (Fig. 4d). Remarkably, biochar derived from agricultural sources can even result in net benefits in certain regions (Fig. 4b). This is primarily driven by the substantial yield enhancement benefits when biochar is utilized for high-yield cereals in Shandong and Henan, or other high-yield crops like sugarcane in Guangxi. While economically valuable crops such as tobacco in Yunnan contribute to reduced costs, Southwest China sees a high average cost of negative emissions from biochar (92.5 $ $t^{-1}CO_2$), as this region's biomass resource predominantly consists of energy crops and grass. For the same reason, the average cost in North China is highest, up to 110.4 $ $t^{-1}CO_2$. Second, the sparser the biomass resource is, the higher the negative emission cost of biochar is. This is because the same pyrolysis plants process less biomass, resulting in a higher unit investment cost (Supplementary Fig. 4). Consequently, the negative emission cost of biochar in Northwest China is high, averaging 100.6 $ $t^{-1}CO_2$ (blue line in Fig. 4d). Finally, yield enhancement co-benefits are more substantial on both coarse-textured (blue pixels, Supplementary Fig. 5) and fine-textured (brown pixels, Supplementary Fig. 5) compared to medium-textured soils. Furthermore,

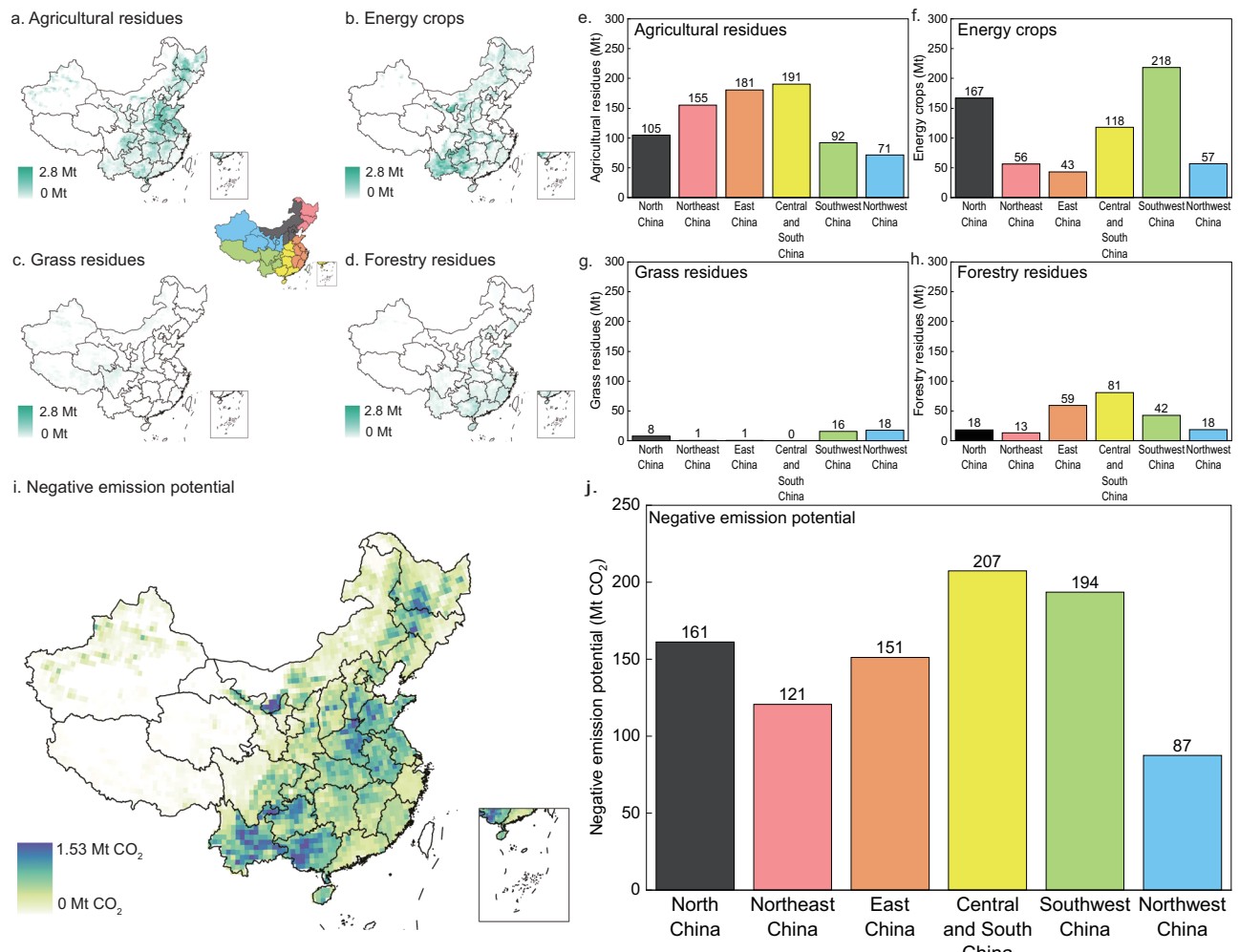

**Fig. 3 | Distribution of biomass feedstocks and negative emission potential under 'Sustainable Technical' scenario. a–d** The distribution of agricultural residues, forestry residues, grass residues, and energy crops on a 0.5° × 0.5° grid. China is divided into six regions: black for North China, red for Northeast China, orange for East China, yellow for Central and South China, green for Southwest China, and blue for Northwest China. Hongkong, Macau, and Taiwan were not included in our analysis. **e–h** Available biomass feedstock in the six regions. **i** Negative emission potential at a 0.5° × 0.5° grid. **j** Negative emission potential in the six regions. Source data are provided as a Source data file. The shapefile of national and provincial boundaries is quoted from the National Catalogue Service For Geographic Information, accessible at www.webmap.cn.

acidic soils (brown pixels in Supplementary Fig. 6) yield greater benefits than alkaline soils[37]. Accordingly, regions in the south might benefit most from yield enhancement.

In summary, Central and South China and East China not only are rich in biomass resources but also have lower costs (yellow and orange lines in Fig. 4d), and could be preferentially selected as pilot areas for biochar application. Taking both cost and potential into consideration, Guangxi Province and Henan Province in Central and South China, and Shandong Province in East China could be priority regions for pilot deployment of biochar. Moreover, pilots in these regions could start with collection of agroforestry residues owing to the associated low cost of producing biochar.

## Discussion

As the need for negative emissions intensifies in the pursuit of rigorous climate targets, it is imperative to investigate the alternative options to premature NETs. The case for biochar, which has over a decade of practical groundwork in China, is particularly compelling. Our study underscores not only the near-term opportunities but also the sustainable prospects of biochar as an established NET in attaining China's carbon neutrality target. We find that biochar presents considerable negative emission potential within China, with the current technical and sustainable technical negative emission potential being 0.43 and 0.92 Gt $CO_2$ per annum, respectively. The average net cost of biochar production stands at around 90 $ $t^{-1}CO_2$, ranging from 60 $ $t^{-1}CO_2$ of forestry-residue-based biochar to 144 $ $t^{-1}CO_2$ of grass-residue-based biochar. Our spatially explicit analysis highlights that region with high potential and low-cost negative emissions largely coincide, primarily in East China and Central and South China. Remarkably, a few areas have the potential to achieve positive returns due to high crop yields or crop value. By offering estimations of the negative emission potential and the economics of biochar, our study can provide recommendations for structured deployment and graded integration of biochar, and provide regional information for the integration of biochar technology into IAMs.

In most regions, although the negative emission cost of biochar is lower than other NETs, it remains higher than the carbon prices in the Chinese national carbon market, making it challenging to promote biochar applications in the near term through offset mechanisms. Yet, promising initiatives have been seen in the United States, Finland, and beyond, where organizations have established voluntary carbon removal platforms that incorporate biochar and have started to explore validation and monitoring methodologies[52]. These undertakings provide valuable insights for implementing biochar

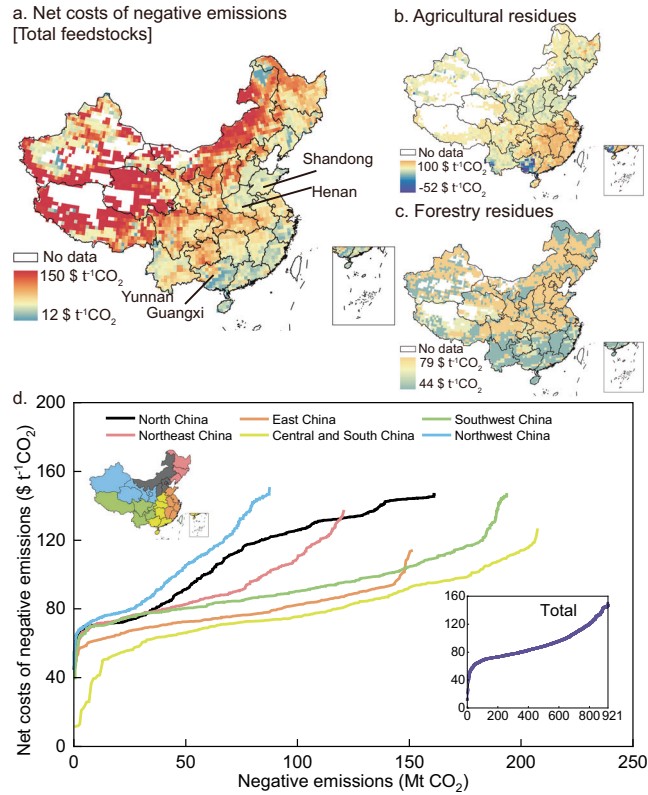

a. Net costs of negative emissions [Total feedstocks]

b. Agricultural residues

c. Forestry residues

d.

**Fig. 4 | Distribution of net cost of negative emissions under 'Sustainable Technical Potential' scenario. a–c** Net cost of negative emissions on a 0.5° × 0.5° grid. Costs are calculated as the weighted average of various feedstocks within the same grid. Note that these net cost estimates differ from those in Fig. 3 owing to the application of site-specific soil conditions and biomass resource in the spatial analysis, whereas the values in Fig. 3 are based on national averages. **d** Net cost curve of negative emissions for various regions. Source data are provided as a Source data file. The shapefile of national and provincial boundaries is quoted from the National Catalogue Service For Geographic Information, accessible at www. webmap.cn.

technology in China. Biochar was included in the 2019 Refinement to the 2006 IPCC Guidelines for National Greenhouse Gas Inventories[53]. However, the development of transparent and locally adapted accounting methods is still in progress and requires further exploration. In order to achieve more ambitious net emission reduction targets, it is imperative to incorporate biochar into national emission trading schemes and Article 6 of the Paris Agreement. This would facilitate the more extensive adoption and effective utilization of biochar technologies for negative emissions.

While this study employs a robust framework and incorporates data from the latest experimental literature and pilot projects, the potential of biochar remains subject to considerable uncertainty. This uncertainty primarily originates from several factors: availability of biomass resources, properties of various biochar types, pyrolysis techniques and conditions, and the impact of biochar application on crop yield. To evaluate this uncertainty, we conduct a Monte Carlo analysis (refer to Supplementary Note 7). The mean estimate of biochar's negative emission potential is 1.07 $GtCO_2$ $yr^{-1}$ under the sustainable technical scenarios, ranging from 0.68 to 1.46 $GtCO_2$ $yr^{-1}$. The mean negative emission cost of biochar is 92 \$ $t^{-1}CO_2$, with a range from −13 to 197\$ $t^{-1}CO_2$. Of all the parameters, those related to by-product income and feedstock purchasing costs are the most sensitive, with the potential to alter biochar cost by approximately 100% (Supplementary Figs. 13 and 14). Furthermore, the negative emission potential of biochar does not include emission reductions attributable to bioenergy production to offset fossil fuel emissions, and avoidance

of soil emissions of methane and nitrous oxide. Both are conceptually different from the negative emission potential but are important factors to consider when assessing the life cycle emissions of biochar. Results indicate that the mitigation potential is 1.5 times greater than the negative emission potential, which are displayed in Supplementary Note 4 and 6. In particular, the negative emission potential and mitigation potential of agricultural residues under the sustainable technical scenario amount to 0.42 Gt $CO_2$ $yr^{-1}$ and 0.76 Gt $CO_2eq$ $yr^{-1}$, exceeding the estimates in existing studies (0.05-0.7 Gt $CO_2eq$ $yr^{-1}$)[31–33], mainly due to our consideration of a wider range of crop types. The explosive effect of soil priming caused by the mineralization of native organic matter due to biochar has also received attention but remains highly debated, as discussed in Supplementary Note 4.

Our study has some limitations. First, our analysis does not encompass all types of biomass resources. We focus on specific feedstocks, leaving out others like livestock manures[29], which have proven potential for biochar production. Moreover, the biochar co-production technology we employed, while economically viable, does not prioritize biochar yield maximization. As carbon budgets become more restrictive, the balance between negative emissions and energy value in biochar production[54] warrants careful consideration. These elements might lead to a potential underestimation of negative emission capacities in our study. Second, our research does not fully account for dynamic influences such as technological advancements, economies of scale, and evolving carbon prices, which could all affect biochar's economic viability and potentially lower its future cost. Lastly, our analysis does not fully address the environmental and socio-economic trade-offs associated with the large-scale deployment of biochar. Increasing biomass demand could potentially result in emissions stemming from land-use changes, intensified competition with food production, and a decline in biodiversity[55]. Simultaneously, the application of biochar can also offer additional co-benefits, such as enhanced heavy metal adsorption in soils[21]. Despite these limitations, our key finding remains that biochar is a promising NET option for achieving carbon neutrality goals in China and should be included in the climate change mitigation toolbox. Future research could aim to explore these potential trade-offs and co-benefits, offering a more holistic understanding of biochar's role in climate change mitigation[27,56,57].

## Methods

### Overall approach

We estimated the negative emission potential and economics of biochar in China using the following steps. First, we calculated the biomass feedstock of 16 types of agricultural residues, 10 types of forestry residues, grass residues, and potential energy crops in China based on statistical and spatial data, which were then aggerated on a 0.5° × 0.5° grid. Then, we developed three scenarios that present maximum theoretical potential, sustainable technical potential, and current technical potential of available biomass feedstocks, respectively. Second, we estimated the negative emission potential of the biochar based on a unified empirical framework distinguishing property of biochar derived from various types of biomass feedstocks. Third, incorporating data from latest experiments in literature and pilot projects, we calculated the economics of biochar and constructed the supply curve of biochar. Finally, considering factors such as soil texture and pH level that might influence the effectiveness of biochar application, we conducted spatially explicit analysis of the negative emission potential and economics of biochar, and identified the locations suitable for biochar deployment. The framework of our approach has been shown in Supplementary Fig. 10.

### Agricultural residues

With reference to Nie et al.[40] and Zhang et al.[58], the 'residue-to-product ratio' method was adopted in our study for its accuracy in determining

the spatial distribution of crops, which incorporated the use of spatial data on crop types, the crop-specific residue-to-product ratio, and the calibration of the result based on national statistics. Here, the residues of 16 types of agricultural crops were considered. Spatial data on various types of crops were obtained from the Harvard Dataverse 2010 with 10-km resolution[59], which were then mapped and calibrated to the crop yields taken from the 2018 China Statistical Yearbook. Relevant formulas and data can be found in Supplementary Note 1.

#### Forestry residues
To clarify the available forestry residues and their spatial distribution, we started from the statistics on planted areas and production at 2018 level[60,61], and adopted method developed by Fu et al. to calculate 10 types of available forestry residues using processing and trimming coefficients of various types of forestry residues[62]. Then, available forestry residues were assigned to grids based on land use type and gridded Net Primary Productivity (NPP) in 2018. The spatial data of wooded and other wooded land were selected from the Resource and Environment Science and Data Center (RESDC) with resolution of 1 km × 1 km[63]. NPP at the 2018 level was obtained from the Moderate-resolution Imaging Spectroradiometer (MODIS) product-MOD17A3HGF.006-with resolution of 500 m × 500 m[64]. The formula and data sources can be found in Supplementary Note 1.

#### Grass residues
First, we sourced statistical data on available grass residues from natural grasslands across the country from the 2018 Annual Report on China's Forestry and Grassland Development[65]. Then, grass residues was assigned to each grid based on NPP that was obtained from MOD17A3HGF.006[64] and spatial land use type data of high-, middle-, and low-coverage grassland were accessed from RESDC[63]. Finally, feedstocks in National Nature Reserves (NNRs) were excluded. The formula used for calculation of grass residues can be seen in Supplementary Note 1.

#### Energy crops
Planting sites and production of energy crops were determined based on the area of marginal land, crop type, and corresponding yields. In this study, marginal lands refer to shrub land, the intertidal zone, bottomland, and unused land that includes sandy, Gobi, saline, marshland, bare land, and bare rocky land. These areas were identified on a 1 km × 1 km grid of land use type[63] with NNRs excluded. C4 plants-sweet sorghum, switchgrass, and miscanthus-were chosen as potential energy crops to be planted in the future. Suitable planting sites for each energy crop were determined based on environmental conditions that included temperature, slope, and precipitation. Yields of each energy crop at the provincial level were adopted from earlier studies[44,46,66]. Finally, maps of the marginal land and three types of energy crops with information of suitable planting sites and potential yields were overlain to determine the best technology potential for individual grids. Consequently, switchgrass was eliminated because of the relatively low yield. The relevant maps, data, and data sources can be found in Supplementary Note 1.

#### Scenarios development on available biomass feedstocks
In this study, we developed three scenarios based on different assumptions on biomass availability: the maximum theoretical potential, sustainable technical potential, and current technical potential. The 'Maximum Theoretical Potential' scenario represents the maximum amount of attainable biomass feedstocks, premised on the assumption that all biomass resources can be harvested and not used for other purposes. The 'Current Technical Potential' scenario signifies the feedstocks available within the constraints of current technologies and practices, with the assumption that only a fraction of

agricultural and forestry residues can be collected–specifically, 88% of agricultural residues[67] and 28% of forestry residues[62] based on the current state of affairs. Contrastingly, the 'Sustainable Technical Potential' scenario falls between the maximum theoretical and current technical potentials. This scenario considers the maximum theoretical potential reduced by the biomass required for livestock and traditional fuels while preserving ecological balance. Here, it is assumed that 95% of agricultural residues and 80% of forestry residues can be collected, after which essential uses are deducted[18]. For regions where theoretical livestock carrying capacity on grasslands is lower than the actual livestock load, no available grass residues were assumed to be harnessed for biochar (refer to Supplementary Note 1 for detailed calculations). The provision for energy crops on marginal land considers unused land and shrub land. The intertidal zone and bottomland are excluded to prioritize ecological conservation. The yield of dedicated energy crops was discounted based on soil quality data from the Harmonized World Soil Database v1.2[68] (Supplementary Note 1).

#### Slow pyrolysis
We assumed the deployment of pyrolysis plants at the center of each grid, producing both biochar and syngas. To standardize the output derived from varying types of biomass, we adopted a unified accounting framework developed by Woolf et al.[39], with the empirical foundation rooted in the existing literature. We used the physicochemical properties of various biomass types as inputs, and sets the pyrolysis temperature at 550 °C for this study. Further, we adopted the biomass gasification technology that co-produces biochar and syngas, as generalized in pilot projects and experimental literature. Biochar's heating value was calculated based on the empirical analyses conducted by Qian et al.[69] Syngas production was calculated by following energy balance and was subsequently used for industrial steam generation. Relevant formulas and data can be found in Supplementary Note 2.

#### Negative emission potential
In this study, the negative emission potential of biochar refers to the $CO_2$ fixed in biomass from the atmosphere through photosynthesis, and then transferred and permanently preserved in biochar. The value was determined based on the quantity of available biomass feedstocks, the weight conversion rate from feedstock to biochar, the carbon content of biochar, and the permanence rate of biochar during 100 years, which were calculated based on accounting framework developed by Woolf et al.[39] and physicochemical composition of crops planted in China. Relevant formulas and data can be found in Supplementary Notes 2 and 3.

#### Yield increasing
We calculated the benefits of yield improvement brought by biochar at optimal application rate by multiplying gridded crop yield[59], crop prices, and the rate of yield increase[31]. Subsequently, we computed the benefits brought by biochar at actual application rate, which was determined by the crop yields and the weight conversion rate from biomass to biochar, using the ratio of the actual to the optimal application rate. For conservative estimation purposes, we set the optimal biochar application rate at 20 t ha$^{-1}$[29]. Relevant formulas and data can be found in Supplementary Note 4.

#### Economic analysis
Cost-benefit analysis was adopted to analyze the economics of biochar, which is one of the commonly used financial assessment method to evaluate the project value by comparing the costs and benefits. Here, the system boundaries were defined as feedstock purchasing, transportation, storage, pyrolysis, biochar transport, application, and effectiveness on crops. It was assumed that feedstock would be transported to a pyrolysis plant located in the center of each grid, while biochar would be returned to the fields in which

the feedstock was harvested. First, net present value (NPV) of 20-year project of pyrolysis plant was calculated. NPV includes the initial investment, annual cash inflows and outflows. Annual cash inflows consisted of increasing yield income and syngas income calculated based on the price and production. Annual cash outflows consisted of the cost of feedstock purchasing, feedstock storage, operation and maintenance, transport of biomass and biochar, and biochar application. Then, the cost of negative emission can be defined as the initial investment and cash outflows apportioned to each unit of $CO_2$ captured in biochar during the whole period. The net cost of negative emission can be considered as the opposite of the NPV apportioned to each unit of $CO_2$. Relevant formulas and data can be found in Supplementary Note 5.

### Spatially explicit analysis

We performed spatially explicit analysis of the negative emission potential and economics. In addition to clarifying the patterns of feedstock distribution, soil texture, PH and biomass abundance in different regions were considered. First, data on soil texture and PH level were adopted from the Harmonized World Soil Database v1.2[68]. With reference to Singh et al.[36] and Wang et al.[37], biochar applied to soil with either coarse and fine texture or acidity was assumed more effective in improving crop yields, as shown in Supplementary Note 4. Furthermore, biomass abundance influenced the investment costs allocated to each pyrolyzed feedstock unit, that is, investing in pyrolysis plants in areas with low biomass was considered less cost-effective, as shown in Supplementary Fig. 4.

### Uncertainty analysis

Using Monte Carlo simulation, we performed uncertainty analysis on the negative emission potential and economics. Random values were generated according to the triangular distribution and normal distribution. We reported uncertainty as a range after 10,000 iterations. We also performed sensitivity analysis on key parameters that might influence the negative emission potential and economics. Further details can be found in Supplementary Note 7.

### Reporting summary

Further information on research design is available in the Nature Portfolio Reporting Summary linked to this article.

## Data availability

Source data have been deposited in Zenodo[70] and GitHub [https://github.com/DXDX97/Biochar_code_and_data]. The data that support the main findings of this study are available in Supplementary Tables 1–13. Raw data on crop spatial distribution, soil, land use type, and NPP used in this study are available in Harvard Dataverse[59], Harmonized World Soil Database v1.2[68], RESDC[63], and NASA[64], respectively. Other data are available from the corresponding author upon reasonable request. Source data are provided with this paper.

## Code availability

The code used to perform the data analysis is available on Zenodo[70] and GitHub [https://github.com/DXDX97/Biochar_code_and_data].

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

## Acknowledgements

We acknowledge the financial support of National Key R&D Program of China (No. 2022YFE0209200), the National Natural Science Foundation of China (72140003,71673162), and Tsinghua University Initiative Scientific Research Program.

## Author contributions

X.D. and F.T. conceived the idea for the work. F.T. guided this study. X.D. collected the data, built the research methodology and performed all calculations. F.T. and M.C. gave important guidance on the framework development. Z.D. gave important guidance on the data collection from literatures and pilot projects. R.L. and P.W. offered support and guidance on the calculation of biomass feedstocks. X.D., F.T., M.C., Z.D., and B.W. discussed the results and contributed to writing the paper.

## Competing interests

The authors declare no competing interests.
