## [Peer Review File · Nature Communications]

REVIEWER COMMENTS

Reviewer #1 (Remarks to the Author):

The study addresses a highly relevant topic and aims to combine state-of-the-art knowledge on biochar application for CDR and further co-benefits in one comprehensive analysis. The assessment could thus be a particularly valuable contribution to both, the scientific progress and political decision-making. The manuscript will thus be highly suitable for publication in Nature Communications, if certain adjustments were made to the framing (e.g. scenario story lines) and if a more solid foundation for the assumptions (e.g. residue availability, biochar parameters, yield increases) was established. This would most likely require new calculations as I expect the current results to (1) drastically overestimate the potential, mostly due to extremely ambitious assumptions for feedstock availability and (2) underestimate the costs, because of unrealistic assumptions on the benefits from by-products and yield increases.

To enhance the quality and validity of the work to an appropriate level, I would recommend to (1) address different pathways and uncertainties in scenarios rather than presenting one result of selected (and mostly highly ambitious/unrealistic) assumptions and (2) revisit the literature for some assumptions and allow transparency towards uncertainty.

Please find my recommendations and comments clustered in the sections below:

Feedstock availability

- The results are presented in a way that suggests equal current availability of feedstocks, yet, while agricultural residues on cropland are (more or less) available today, the large-scale implementation of biomass plantations would require some time.

- Using grassland “residues” is not really a convincing concept. Is there any evidence from current practice? For both, grazing and mowing, you would not really generate a significant mass of residues. Also, in terms of management and C allocation, it is a significant difference if the area is grazed or mowed. Thus, the economic viability would significantly differ between grassland that is mowed (approach maybe feasible) and grazing areas (highly questionable if feasible). However, this questionable source is making up a significant contribution to your overall potential.

Further, the residue-to-product ratio seems to be unrealistic (too high). The references don't support this value, as Turner et al. only give a rough estimate of harvestable NPP over storage capacities. These numbers do not qualify for deriving a residue-to-product. Also, it is unclear how the numbers of Nie et al. are used. Their formula gives a fraction of approximately 0.1.

- The marginal land that is assumed for bioenergy crop production includes areas that are drastically unsuitable, like intertidal zone, Gobi, saline and bare rocky land. Why should anything grow there? Given

the unsuitable land you mention, the variables assessing the suitability are insufficient. Some areas you consider are regularly flooded or lack appropriate soil layers. Assessed conditions are not aligned with the unsuitable conditions in “marginal” regions.

Assuming the yields of the province as a proxy for biomass yields on marginal land is not an appropriate representation. The yields will be varying across one province, given the diverse conditions, especially when considering the highly unsuitable “marginal” areas. Why would marginal land produce the same yields as other areas of the province/trial plots/highly productive conditions? They are marginal for a reason.

Revenues from by-products:

- The R&D and investments necessary for this revenue that contributes significantly to the relatively low price of biochar is not in line with the argument of a mature and ready-to-use technology (opposed to BECCS or DACCS)
- Especially with the R&D required for the bio-oil and syngas revenue, the price of 132–247 \$/tCO₂ is much more likely in the near future. The option for by-product sale (as a pathway of low technological readiness) should be addressed in scenarios or separated in some other way.
- When assessing the bioenergy production to offset fossil fuel emissions it is not addressed that BECCS would be far more effective in energy supply (+ C sequestration) and is thus more likely to be implemented. Especially the use of bio-oil and syngas would need far more R&D.
- Why is the syngas not used for powering the process as usually assumed/common practice?

Biochar parameters:

- In general, the assessment would benefit from parameters collected from meta-analyses or frameworks rather than single studies. For example, Woolf et al. (2021) provide a framework for calculating biochar yields and carbon contents based on different HTT and feedstocks.
- The weight conversion rate for rice is surprisingly high for the low lignin content of rice straw and is expected to be much lower, i.e. around 23% for 550°C HHT (see Woolf et al 2021). Also, the C content of rice-based biochar seems to be particularly high and is expected to be much lower, i.e. around 0.48 (see Woolf et al. 2021).
- The 10% decay rate for the time period of 10% is rather optimistic. How does that relate to other assumptions, i.e. Woolf et al (2021).
- Do you use an application rate of 2t/ha or 20t/ha? What rate was used for generating the results? And do you somehow account for larger areas that can be treated with biochar in the 2t/ha scenario?

Yield increases:

- Overall, I assume the effect of yield increases to be highly overrated in this assessment.
- In general, the numbers assumed for yield increases are particularly high, compared to other meta-analyses (i.e. Ye et al. (2020), Jeffery et al (2017)). It is not discussed how these high numbers relate to these former findings. It is unclear which reference was used for the yield increase of each crop. Further, the references used are rather unsuitable. Major et al. (2019) analyse very specific conditions for maize that are not applicable in this study. Wang et al (2019) evaluate the effect of co-composted biochar, which includes a significant effect of the fertilization from the compost. The effect is not separated in the meta-analysis because not all studies include fertilized controls. However, the Chinese agricultural includes vast areas of high fertilizer application rates. These numbers are thus not useful here.
- Also, it is not clear from the description in the SI whether you use average yields for all areas of one crop, i.e. wheat, or whether you use disaggregated values, like the ones you use for the residue calculations.

Structure:

- Figure 5: all maps need letters for identification, i.e. a), b), c), d)
 - Methods: The methods section is partly lacking relevant information. For example, in the paragraph of forest residues, you mention statistics and gridded data without referencing it. Also, for the coefficients it would be good to know how they are derived or if they are literature-based. The same is true for the residue-to-product ratio of grassland. In the part on slow pyrolysis, you mention that temperature, duration, and equipment are important, yet give no information on i.e. what temperature you assume. Also, it is not clear, whether you assume the produced syngas to be used for power generation or if another source is used.
- In the section on negative emission potential, you need to at least address which approach or database you follow for the biochar yield and carbon content in the char – this information is crucial for the overall NE potential and currently not described sufficiently. Also, you need to clarify the period of time here.
- Also, you need to describe how you quantify the yield increases in the Methods section.

Comments on understanding:

p.8 : “For example, the share of agroforestry residue is 80% in East China and 66% in Central and South China, with low average costs of 66.7 and 68.1 \$/tCO₂, respectively (Fig. 5c).”

- What is the 80% referring to? Where do I find 5c?

p.9: “To date, no biochar offset credit projects have been implemented, mainly because of the lack of suitable baselines and monitoring methodologies.”

- What about the voluntary market?

p.10: "Additionally, further strengths and drawbacks of biochar implementation should be examined carefully to minimize the trade-offs with key sustainable development goals, e.g., heavy metal sorption, loss of biodiversity⁴³, and land use change emissions."

- Loss of biodiversity is a trade-off, while heavy metal sorption is a co-benefit. Also, it needs to be clarified with what these trade-offs are associated, i.e. feedstock generation or biochar application to soils

Reviewer #2 (Remarks to the Author):

The manuscript describes the potential negative emission potential, economics and spatial deployment of biochar applications in China. Although the topic is within the scope of the journal, the current version of the manuscript does not present a clear novelty compared to other papers on the field. Please, find some suggestions for improvement below:

1. Research questions could be better elucidated. Apart from the "limited to research on agricultural residues", are there limitations in the current assessment methods or data?
2. Lines 59-61 "In fact, the negative emission potential and cost of biochar derived from various feedstocks differ greatly owing to their disparate properties, namely, porosity, surface area, and carbon content." The chapter on cost of negative emissions from biochar only focuses on carbon content. How do the other two factors affect economics without any discussion?
3. For the cost analysis of negative emissions of biochar, please include cost breakdown and financial assessment, also suggest adding the storage cost of biomass.
4. In the chapter on spatial analysis of biochar potential, only soil texture is mentioned as an influencing factor on crop yield, soil pH is also a key factor on crop yield and is suggested to be added.
5. The paper mentions that biochar has negative emission potential, which is consistent with China's goal of achieving carbon neutrality by 2060. Could a dynamic assessment model be used to simulate the negative emission potential of biochar from now until 2060?
6. What are the limitations of the proposed analysis? Close out in either the methodology or discussion. Make sure to discuss how this would affect the results; then highlight again in the conclusion and finger towards further research you or other could undertake to improve.

7. What are the potential sources of error in this analysis, and how can the underlying assumptions be improved?

Point-by-point Response to Review Comments on Biochar

Reviewer #1 (Remarks to the Author):

The study addresses a highly relevant topic and aims to combine state-of-the-art knowledge on biochar application for CDR and further co-benefits in one comprehensive analysis. The assessment could thus be a particularly valuable contribution to both, the scientific progress and political decision-making. The manuscript will thus be highly suitable for publication in Nature Communications, if certain adjustments were made to the framing (e.g. scenario story lines) and if a more solid foundation for the assumptions (e.g. residue availability, biochar parameters, yield increases) was established.

This would most likely require new calculations as I expect the current results to (1) drastically overestimate the potential, mostly due to extremely ambitious assumptions for feedstock availability and (2) underestimate the costs, because of unrealistic assumptions on the benefits from by-products and yield increases. To enhance the quality and validity of the work to an appropriate level, I would recommend to (1) address different pathways and uncertainties in scenarios rather than presenting one result of selected (and mostly highly ambitious/unrealistic) assumptions and (2) revisit the literature for some assumptions and allow transparency towards uncertainty. Please find my recommendations and comments clustered in the sections below:

Dear reviewer, we sincerely appreciate your recognition of the paper's topic and your thorough review, which provided valuable professional insights. We have carefully considered and addressed each of your comments, resulting in significant revisions that enhance the quality and reliability of our research.

To address the issue of potential overestimation (1), we conducted a comprehensive re-evaluation of available biomass feedstocks. As a result, we have introduced three distinct scenarios: current technical potential, sustainable technical potential, and maximum theoretical potential. These scenarios are further explained in response to comment 1, providing a more nuanced and appropriate assessment of the biochar potential in China.

To tackle the concern regarding cost underestimation (2), we conducted surveys with six biochar manufacturers located in different provinces to gather cost related data. Additionally, we carefully incorporated relevant experimental data from existing literature. By combining practical project insights with theoretical considerations, we have significantly enhanced the robustness and validity of our results, ensuring they are aligned with the reality of biochar production in China. This discussion is further elaborated in response 2.

Furthermore, we have incorporated recent meta-analysis publications on the effects of biochar application on crop yields in China in 2023. This data informed our

determination of parameters related to the benefits of crop yield enhancement for different crops. By considering the latest research, we have improved the accuracy and relevance of our assessments, as described in response 4.

Regarding the determination and modifications of other parameters related to biochar and the overall framework of the paper, we have provided comprehensive explanations in response to comment 3 and response to comment 5, respectively. These detailed responses offer clarity and insight into our research methodology and decision-making process.

Once again, we express our gratitude for your meticulous review. Your feedback has played a crucial role in refining our work and strengthening its scientific rigor. We believe that the revisions we have made, as outlined in the respective responses, significantly enhance the quality and validity of our study.

1. Feedstock availability

1.1. The results are presented in a way that suggests equal current availability of feedstocks, yet, while agricultural residues on cropland are (more or less) available today, the large-scale implementation of biomass plantations would require some time.

Response: Thank you so much for highlighting the variability of available biomass feedstocks under different scenarios. On one hand, achieving large-scale bioenergy plantations in the near future can be challenging. On the other hand, excluding the potential of future bioenergy plantations may underestimate the negative emission potential of biochar. According to your suggestions regarding exploring different scenarios, we have examined the maximum theoretical potential, sustainable technical potential, and current technical potential of biomass feedstocks. The details of scenarios are provided below:

Table 1 Potentials of available biomass feedstocks under various scenarios.

Scenarios	Maximum theoretical potential	Sustainable technical potential	Current technical potential
Description	Maximum amount of available biomass feedstock	Theoretical potential minus use for livestock, etc., while maintaining ecology	Available biomass resources limited by current technology and practice
Agricultural residues	0.99 Gt/a	0.79 Gt/a	0.73 Gt/a
	100% of 16 types of crop residues	95% of crop residues those are not used as feed, substrate and raw material	88% of residues those are not used as feed, substrate and raw material
Forestry residues	0.29 Gt/a	0.23 Gt/a	0.04 Gt/a
	100% of 10 types of forestry residues	80% of 10 types of forestry residues	28% of 10 types of forestry residues
Grassland residues	0.29 Gt/a	0.04 Gt/a	0.00 Gt/a
	100% of hay production in natural grasslands	hay production in natural grasslands not used as feed	Not available under current technology
	0.86 Gt/a	0.66 Gt/a	0.00 Gt/a

Energy crops	Maximum production potential of dedicated energy crops; Marginal land refers to shrub land, the intertidal zone, bottomland, and unused land	Production potential of dedicated energy crops discounted by land productivity; Marginal land refers to unused land and shrub land, where the national special shrubland is excluded	Not available under current technology
Total biomass feedstocks	2.43 Gt/a	1.73Gt/a	0.77 Gt/a

The maximum theoretical potential represents the maximum amount of biomass feedstock available, assuming all biomass resources can be harvested without any competing uses. This scenario is optimistic for biochar application, while it is possible to achieve a high biochar potential if biochar has more advantages (e.g., economical competitive) compared to other technologies.

The current technical potential represents the biomass resources available based on existing technologies and practices. It assumes that only agricultural and forestry residues can be partially collected, while deducting biomass for other purposes. In terms of agricultural residues, in 2021, the comprehensive utilization rate of straw in China was 88.1%, with 20% of utilized feedstocks applying for feed, raw materials, and substrates. Therefore, we assume that 88% of agricultural residues can be collected, and then deducted by basic uses. It is worth noting that a significant portion of straw utilization is for returning to the fields, accounting for approximately 50%. Given that biochar itself can contribute to soil sustainability through its application as soil amendment, we have not excluded this portion. For forestry residues, we assume that 28% can be collected, which is according to the current situation.

The sustainable technical potential lies between theoretical potential and current technical potential. It represents the theoretical potential minus the use of biomass for livestock and traditional fuels while maintaining basic ecological balance. Meanwhile, technology has surpassed current levels, as reflected in the increased collection rates of agroforestry residues and grassland, and the possibility of large-scale plantation of dedicated energy crops. In particular, agricultural residues can be 95% collected, but deduction is also made for uses of feed, raw materials, and substrates. Available grassland residues are merely 15% of the theoretical potential after deducting hey for livestock. Marginal land for energy crops only considers unused land and shrubland that excludes area under the National Special Shrublands, which is mainly located in arid and ecologically vulnerable areas. Additionally, intertidal zone and bottomland are excluded considering ecological conservation. We have provided detailed explanations regarding the sustainable scenarios for grassland and energy crops in response 1.2 and 1.3.

We have made the following modifications to the paper:

Firstly, we have added this table in Supplementary Table 8.

Secondly, we have revised almost entire Result 1 in main text to provide detailed descriptions of the three scenarios, their corresponding biomass resources, and the potential of negative emissions.

Thirdly, we have added Scenario Development in the Method section, and provided detailed descriptions of assumptions, formulas, and parameters in the Supplementary Information Note 1.

1.2. Using grassland “residues” is not really a convincing concept.

1) Is there any evidence from current practice?

Response: Thank you for your comment regarding the utilization of grassland residues for biochar production. We appreciate your inquiry, and based on the following reasons, we firmly believe that there are indeed opportunities to employ grassland residues in this context, albeit with limitations compared to other biomass resources.

Firstly, recent local reports¹ indicate the active engagement of farmers in harvesting alfalfa grass for sale to factories. This observation suggests that if grass were to become economically viable, grassland residues could be utilized for various purposes. Although specific instances of grassland residues being employed for biochar production have not been encountered yet, we acknowledge the potential for such endeavors in the future.

Furthermore, the availability of grassland residues is primarily influenced by the mismatch between grass supply and demand, both in terms of quantity and quality. It is crucial to address the existing spatial mismatch within China's livestock industry. Overgrazing has rendered significant grasslands unsuitable for alternative uses, while other grasslands possess a theoretical livestock carrying capacity that exceeds their actual livestock load. This discrepancy presents an opportunity for strategic planning and resource allocation for alternative purposes.

In addition, there is a notable mismatch in terms of forage quality, which emphasizes the need for alternative applications of grassland residues. Livestock preference for high-quality grass, coupled with regions characterized by unfavorable climatic and soil conditions, results in a disparity between the available low-quality grass supply and the demand for suitable feed. By utilizing such low-quality grass for biochar or bioenergy production, we can potentially maximize its value and address the supply-demand imbalance.

However, it is essential to consider the context of sustainability and the competition with the livestock industry. The availability of grassland residues for biochar production is severely limited. Based on your comments, we revised our methodology to estimate the potential of grassland residues, our new analysis indicates that the estimated quantity of grassland residues suitable for this purpose is a mere 43 Mt, as elaborated in sections 2) and 3) below. It is imperative to acknowledge these limitations when evaluating the potential of grassland residues as a biochar resource.

Reference:

1. The alfalfa grass in Jia County has a good harvest, YNET (in Chinese), 2023, <https://baijiahao.baidu.com/s?id=1767006159067214219&wfr=spider&for=pc>

2) Further, the residue-to-product ratio seems to be unrealistic (too high). The references don't support this value, as Turner et al. only give a rough estimate of harvestable NPP over storage capacities. These numbers do not qualify for deriving a residue-to-product. Also, it is unclear how the numbers of Nie et al. are used. Their formula gives a fraction of approximately 0.1.

Response: Thanks for your question regarding the availability and utilization pathways of grassland residues. To demonstrate our modification process in a better way, we address the 'residue-to-product ratio' first, followed by the issue of mowing and grazing.

Instead of using one general and unclear residue-to-product ratio as what we did before, we reorganized the literature to clarify the relationship between Net Primary Productivity (NPP) and available grassland biomass (maximum theoretical potential). Firstly, similar to the previous version, we have obtained the 2018 median NPP data for China from MODIS products and filtered the data on grassland based on land use type grid data. Subsequently, we calculated the available grassland in dry weight using the following formula:

$$\text{Grass residues} = \frac{\text{NPP}}{0.45 \text{g} \cdot \text{gC}^{-1} \cdot (1+r)}$$

where r represents the ratio between belowground and aboveground biomass, of which the value is 3.81¹. This calculation yields a grassland production estimate of 391 Mt.

Subsequently, we compared this estimate with existing studies, e.g. the study of Nie et al.², the available biomass is calculated as: Grass residues = $\text{NPP} * \frac{0.2}{0.5}$, resulting in 0.38Gt of grass residues in 2015, which is similar to our estimates.

Then, we compared our estimate to statistical data. According to the 2018 Annual Report on China's Forestry and Grassland Development, the total fresh grass production on natural grasslands nationwide was reported as 1,099 Mt, which is equivalent to approximately 339 Mt of dry grass. The statistical data from the report reflects various grassland types by diverging coefficient values (r), which is more suitable than adopting a single value r .

Therefore, we ultimately chose not to use the method of residue-to-product ratio, but instead opted for the method that using the statistical data as the total amount, 339 Mt, and then downscaled it based on the NPP data as a weighting factor while excluding National Nature Reserve. This process yielded a final estimate of 288 Mt as the

theoretical potential of available grassland residues. We revised our methodology to estimate the potential of grassland residual in Supplementary Information Note 1.3.

References

1. WANG Qi, WU Chengyong, CHEN Kelong, et al. Estimating grassland yield and carrying capacity in Qinghai Lake Basin based on MODIS NPP data. *Ecological Science (in Chinese)*, 2019, 38(4): 178-185.
2. Nie, Y. et al. Spatial distribution of usable biomass feedstock and technical bioenergy potential in China. *GCB Bioenergy* 12, 54–70 (2020).

3) For both, grazing and mowing, you would not really generate a significant mass of residues. Also, in terms of management and C allocation, it is a significant difference if the area is grazed or mowed. Thus, the economic viability would significantly differ between grassland that is mowed (approach may be feasible) and grazing areas (highly questionable if feasible). However, this questionable source is making up a significant contribution to your overall potential.

Response: Thank you for your insightful comment regarding the feasibility of utilizing grassland residues for biochar production. We appreciate your attention to the differences between grazing and mowing in terms of residue generation and management implications.

We revised our methodology to estimate the potential of grassland residual in Supplementary Information Note 1.3. To provide a clearer understanding of our research methodology, we would like to elaborate on the data and calculations used to estimate the potential of grassland residues.

Initially, we obtained data from the 2018 National Forestry and Grassland Development Statistical Bulletin, which reported a total production of fresh grass on natural grasslands in China as 109,942.02 million tons. This is equivalent to approximately 33,930.75 million tons of dry hay, with a carrying capacity of about 26,717.12 million sheep units. It's important to note that we downscaled this statistical data using NPP spatial data and excluded national protected areas to ensure a more accurate representation.

Furthermore, we conducted calculations to determine the actual livestock load. We acquired end-of-year livestock population data for cattle, horses, and sheep from the 2018 China Livestock Yearbook and downscaled the statistical data using the Gridded Livestock of the World (GLW) 2015 livestock spatial data. By applying the grassland carrying capacity, the grass-livestock balance calculation method, and the calculation of the reasonable carrying capacity of natural grasslands, we converted livestock numbers into sheep units. This enabled us to estimate the actual livestock load at the grid scale.

Based on the assumption that an adult sheep weighing 45 kg consumes 1.8 kg of standard dry hay daily, we calculated the actual livestock load and compared it with the

theoretical grassland carrying capacity. If the theoretical quantity of grassland resources is lower than the actual quantity, we assume that there are no grassland residues available for biochar production. However, if the theoretical quantity exceeds the actual quantity, we calculate the difference as the available residue quantity.

It is important to note that our calculation method is relatively conservative. The statistical data we used accounts for only 40%-60% of above-ground biomass as available for carrying capacity calculation in the case of grazing. However, with the future promotion of mowing, the utilization rate can increase up to 80%. Additionally, China is actively promoting the cultivation of high-quality forage crops on arable land, which allows for more rational planning and integrated development of livestock, bioenergy, and negative emission technologies on grasslands.

We hope this clarification provides a comprehensive understanding of our research approach and the factors considered in estimating the availability of grassland residues for biochar production.

1.3. The marginal land that is assumed for bioenergy crop production includes areas that are drastically unsuitable, like intertidal zone, Gobi, saline and bare rocky land.

1) Why should anything grow there?

Response: Thank you for your question. First, the definition of marginal land varies in studies, generally referring to lands with limited climate, slope and soil conditions, making it unsuitable for conventional crop cultivation¹.

However, existing studies also pointed out that ‘the same characteristics that make a site marginal in one location or for one purpose can make land productive in another location or for a different purpose’¹⁻³. Indeed, marginal land is considered crucial to produce second-generation bioenergy crops, specifically lignocellulosic biomass crops. This is because certain energy crops can thrive in harsh environmental conditions. For example, sweet sorghum is a representative non-food energy crop for biofuel production in China due to its high biomass yield, drought and cold resistance, flood tolerance, wide adaptability, and high sugar content. Miscanthus is a perennial C4 plant with efficient photosynthetic pathways. Switchgrass is a perennial grass species native to subtropical and tropical Asia, capable of producing high biomass and surviving in low-quality soils under extreme climatic conditions.

In China, deployment of energy crops should not compete with agricultural land or threaten food security. Therefore, utilizing marginal land for bioenergy production is seen as a key solution to meet both food security and energy demands. Based on existing literature regarding the definition of marginal land in China (as shown in the table below), in the theoretical potential scenario, this study defines marginal land as shrubland, tidal flats, beach land, and unused land. In the sustainable potential scenario, marginal land is defined as shrubland and unused land.

Table 2 The definition of marginal land and results comparison between existing studies and this study

Study	Marginal land	Area (Mhm ²)	Production	Energy crops
this study	Shrub land, the intertidal zone, bottomland, and unused land including sandy, Gobi, saline, marshland, bare land, and bare rocky land.	53	0.86 Gt/a	Miscanthus Sweet sorghum Switchgrass
	Shrub land, and unused land including sandy, Gobi, saline, marshland, bare land, and bare rocky land.	50	0.66 Gt/a	
Zhang et al.,2020 ⁴	Sparse grassland, Shrub land, Sparse Forest land, Moderate coverage grassland, High coverage grassland, Saline-alkali land, Bottomland, Bare land, Intertidal zone	184.9	1.76Gt/yr 0.284Gt/yr	Miscanthus, Switchgrass
Nie et al., 2019 ⁵	Tidal-flat land, sand land, saline-alkali soil land, swampland, bare land, etc. (Precipitation ignored)	91.49–102.09	12.30-20.46 EJ	Sweet sorghum
Jiang et al.,2019 ⁶	Shrub land, sparse forest land, grassland, shoal/bottomland, alkaline land and bare land.	49.65	13.57Mt 0.848Mt ethanol 115PJ	Sweet sorghum
Qin et al.,2018 ⁷	alkaline land, bare land, degraded land, waste land, and idle land	3 - 100		Review
Xue et al.,2016 ⁸	Sparse grassland, Shoal, Bottomland, Sand land, Gobi, Alkaline land, Wetland, Bare land, Bare rock	171.64(margin al land) 7.69(suitable)	0.135Gt/yr 183.9TWh/yr 0.212GtCO ₂ eq/yr	Miscanthus

We have added the table above to SI Table S4. We hope this response addresses your query appropriately.

References:

1. Csikós, N. & Tóth, G. Concepts of agricultural marginal lands and their utilisation: A review. *Agricultural Systems* 204, 103560 (2023).
2. Theory and Identification of Marginal Land and Factors Determining Land Use Change. (2010). doi:10.22004/ag.econ.98203.
3. Dale, V., Kline, K., Wiens, J. & Fargione, J. Biofuels: Implications for Land Use and Biodiversity *Biofuels: Implications for Land Use and Biodiversity*. (2010).
4. Zhang, B., Hastings, A., Clifton-Brown, J. C., Jiang, D. & Faaij, A. P. C. Modeled spatial assessment of biomass productivity and technical potential of *Miscanthus × giganteus*, *Panicum virgatum* L., and *Jatropha* on marginal land in China. *GCB Bioenergy* 12, 328–345 (2020).

5. Nie, Y. et al. Assessment of the potential and distribution of an energy crop at 1-km resolution from 2010 to 2100 in China – The case of sweet sorghum. *Applied Energy* 239, 395–407 (2019).
6. Jiang, D., Hao, M., Fu, J., Liu, K. & Yan, X. Potential bioethanol production from sweet sorghum on marginal land in China. *Journal of Cleaner Production* 220, 225–234 (2019).
7. Qin, Z. et al. Biomass and biofuels in China: Toward bioenergy resource potentials and their impacts on the environment. *Renewable and Sustainable Energy Reviews* 82, 2387–2400 (2018).
8. Xue, S., Lewandowski, I., Wang, X. & Yi, Z. Assessment of the production potentials of *Miscanthus* on marginal land in China. *Renewable and Sustainable Energy Reviews* 54, 932–943 (2016).

2) Given the unsuitable land you mention, the variables assessing the suitability are insufficient. Some areas you consider are regularly flooded or lack appropriate soil layers. Assessed conditions are not aligned with the unsuitable conditions in “marginal” regions. Assuming the yields of the province as a proxy for biomass yields on marginal land is not an appropriate representation. The yields will be varying across one province, given the diverse conditions, especially when considering the highly unsuitable “marginal” areas. Why would marginal land produce the same yields as other areas of the province/trial plots/highly productive conditions? They are marginal for a reason.

Response: Thank you so much for raising this question. As you mentioned, planting sweet sorghum, miscanthus or switchgrass did require suitable climate and soil conditions. To address this issue, firstly, we excluded certain regions based on temperature, slope, and precipitation, as we did in the previous version. At this stage, most of the harsh and unused land conditions has already been eliminated. Secondly, as you pointed out, soil quality is crucial, so we used the soil quality data from Harmonized World Soil Database v1.2¹. We assumed the yield data from existing literature as the optimal yield, and discounted the productivity according to soil quality. The resulting energy crop planting area is 50.5 Mhm², with a yield of 0.66 Gt/a, which is consistent with the range of existing estimates of 3-185 Mhm².

We revised our methodology to estimate the potential of energy crops in Supplementary Information Note 1.4:

‘Besides, in terms of sustainable potential, since the soil quality would have influence on the growth potential of dedicated energy crops, we discounted the crop yield based on the soil quality data from Harmonized World Soil Database v1.2, which includes seven evaluation dimensions such as nutrient availability, with each dimension having 1-7 levels. The description of soil quality said ‘Only classes 1 to 4 are corresponding to an assessment of soil limitations for plant growth. Class 1 is generally rated between 80 and 100% of the growth potential, class 2 between 60 and 80%, class 3 between 40 and 60%, and class 4 less than 40%.’ Therefore, we took the yield data from existing literature as the optimal yield, took the grades with lowest level for each grid, and discounted the productivity (Pt_m) according to the description.’

Reference:

1. Fischer, G. et al. Global Agro-ecological Zones Assessment for Agriculture (GAEZ 2008). IIASA, Laxenburg, Austria and FAO, Rome, Italy. (2008).

2. Revenues from by-products:

2.1. The R&D and investments necessary for this revenue that contributes significantly to the relatively low price of biochar is not in line with the argument of a mature and ready-to-use technology (opposed to BECCS or DACCS). Especially with the R&D required for the bio-oil and syngas revenue, the price of 132–247 \$/tCO₂ is much more likely in the near future. The option for by-product sale (as a pathway of low technological readiness) should be addressed in scenarios or separated in some other way.

Response: Thank you so much for suggestions about by-products. To address the by-product sale, we reviewed a series of experiments from existing studies to determine the type of pyrolysis technology, and then conducted the survey on six biochar pilot projects, which has provided us with in-depth insights into biochar production technologies currently used in practice, as well as the sales channels and profits of corresponding by-products. Given that in our revised version, by-product was assumed to be merely syngas other than both syngas and bio-oil, according to the technology of biomass gasification for biochar and syngas co-production, we did not develop scenarios but address analysis regarding with/without by-products.

We have made the following modifications to the paper:

Firstly, we added Table S11 Parameters of biomass gasification for syngas and biochar co-production technology, and Table S12 Parameters of Pyrolysis Plants from Surveys to Supplementary information.

Secondly, we added description in Supplementary Note 2:

‘Besides biochar, there are also by-products generated during the process of biomass pyrolysis. There are various technologies for biochar production, including carbonization, dry distillation, and gasification with multiple co-production techniques. The by-products may include syngas, bio-oil, wood vinegar, electricity, and others. Correspondingly, the energy efficiency varies significantly depending on the specific technology and process conditions.

To ensure that the identified technology type and the benefits from by-products align closely with reality in China, we conducted both literature research and obtained first-hand data from biochar production facilities, which aimed at facilitating a more accurate and reasonable economic evaluation.

Firstly, we selected the technology of biomass gasification for syngas and biochar co-production. Despite having a relatively lower weight conversion rate of biochar from

biomass compared to other technologies (around 1/5 to 1/4), it has demonstrated significant economic benefits from both biochar and syngas and is considered one of the key industrialization directions for biochar production in China. Some domestic research teams, including Nanjing Agricultural University, Chinese Academy of Agricultural Engineering (Ministry of Agriculture), and Huazhong University of Science and Technology, have been actively involved in this field and performed a series of experiments, as shown in Table S10, based on which, we have determined the energy efficiency, with a median value of 71% (μ , %).

Secondly, our survey on existing biochar pilot projects also supports the feasibility and development potential of biomass gasification for co-production technology, as shown in Table S11. According to the practical data, we identified the types of by-products and their benefits. Given that in biomass gasification technology, gaseous tar and other products were almost combusted to powering the process, by-product benefits solely come from remaining usable syngas. Syngas has some main uses: for self-use to save heating expenses and conversion into industrial steam or electricity. Here, syngas was assumed to be converted into industrial steam, as some of the pilot project did (Table S11). The heat value of medium-temperature and medium-pressure steam commonly used in industries at 400°C and 4 MPa is approximately 3.278 GJ/t, with a price of 250 yuan per ton. The energy efficiency from syngas to steam is 92%. In this way, the benefits of syngas can be calculated.'

Finally, we revised our estimate on benefit from syngas and then made corresponding modification in main text Result 2.

2.2. When assessing the bioenergy production to offset fossil fuel emissions it is not addressed that BECCS would be far more effective in energy supply (+ C sequestration) and is thus more likely to be implemented. Especially the use of bio-oil and syngas would need far more R&D.

Response: We appreciate the suggestion to include a discussion on BECCS, which we added in our Supplementary Information Note 6:

'It is worth noting that the baseline corresponding to the mitigation potential here refers to current situation. Fossil fuel emission offsets, for example, was calculated as the differences between syngas and natural gas to produce the same quantity of industrial steam. Another alternative option of energy production - BECCS - is likely to be used in the future but not considered in this study. The by-product of BECCS can be electricity, which implies a higher energy availability and is more ready-to-use than by-products of pyrolysis. On the other hand, limitations associated with BECCS are obvious, mainly including high carbon capture costs, poorly constructed carbon transport pipelines (especially in China where there is a serious mismatch of biomass sources and basin sinks), and carbon storage that may face leakage or lead to geological risks. Besides, the availability of residues for BECCS might be lower compared to biochar due to the need to maintain soil fertility, and the left residues might also result in soil

GHG emissions. In general, the comparison of biomass-based negative emission technologies needs further investigation.’

Indeed, our ongoing research indicates that considering these limitations, the actual negative emissions potential of BECCS in China drop to no more than 10% of its theoretical potential. From this perspective, biochar is more likely to be ready to use. The comparison between biochar and BECCS from a comprehensive standpoint is necessary to be addressed, and we look forward to furthering investigations in this regard.

2.3. Why is the syngas not used for powering the process as usually assumed/common practice?

Response: Thank you for your clarification. In our revised version, in terms of the biomass gasification for biochar and syngas co-production technology, gasified tar is fed into the combustor for powering the process, while the remaining syngas can be utilized for industrial steam production. The technical details are presented in the supplementary information Note 2.

3. Biochar parameters:

3.1. In general, the assessment would benefit from parameters collected from meta-analyses or frameworks rather than single studies. For example, Woolf et al. (2021) provide a framework for calculating biochar yields and carbon contents based on different HTT and feedstocks.

Response:

We sincerely appreciate your valuable suggestion regarding the use of parameters collected from meta-analyses or frameworks in our assessment. Your insight has guided us in improving the robustness and reliability of our study.

In response to your comment, we have carefully examined the assessment framework proposed by Woolf et al. (2021) and acknowledge its relevance to our research. This framework provides an empirical formula that effectively captures the diverse characteristics of different biomass types for biochar conversion, making it highly suitable for our study.

Based on this valuable reference, we have made necessary modifications to our methodology to align with the specific characteristics of Chinese crops. Specifically, we have replaced the ash content and lignin content data for major crop residues to ensure greater accuracy and applicability within the context of our analysis.

By incorporating the insights provided by Woolf et al. (2021), we have enhanced the scientific rigor and validity of our study. These changes have allowed us to effectively evaluate the biochar conversion process for different biomass types in China, providing

more robust results and conclusions. We have made the modifications to Supplementary information Note 2.

1. Woolf, D. et al. Greenhouse Gas Inventory Model for Biochar Additions to Soil. *Environ. Sci. Technol.* 55, 14795–14805 (2021).

3.2. The weight conversion rate for rice is surprisingly high for the low lignin content of rice straw and is expected to be much lower, i.e. around 23% for 550°C HHT (see Woolf et al 2021). Also, the C content of rice-based biochar seems to be particularly high and is expected to be much lower, i.e. around 0.48 (see Woolf et al. 2021).

Response: Thank you for bringing this to our attention. We have reviewed the literature and made the necessary modifications. Firstly, we obtained the ash content and lignin content of rice straw and rice husk from existing studies through proximate analysis and componential analysis. Subsequently, using Woolf's empirical formula, we calculated the conversion rates and carbon contents of rice husk and rice straw at 550°C. The conversion rate of rice straw was determined to be 22.9%, with a carbon content of 52.9%. The conversion rate of rice husk was found to be 25.2%, with a carbon content of 49.3%. We believe that these updated values, which are specific to rice husk and rice straw in China, better reflect the characteristics of these agricultural residues and contribute to the accuracy of our analysis. We have made the modifications to Supplementary information Note 2.

3.3. The 10% decay rate for the time period of 10% is rather optimistic. How does that relate to other assumptions, i.e. Woolf et al (2021).

Response: Thank you very much for bringing this to our attention. We have taken into account Woolf's article and conducted calculations accordingly. The permanence rate over 100 years is estimated to be approximately 75% in terms of biochar polysized at the temperature of 550 °C . We have made the modifications to Supplementary information Note 3 and the Method section.

3.4. Do you use an application rate of 2t/ha or 20t/ha? What rate was used for generating the results? And do you somehow account for larger areas that can be treated with biochar in the 2t/ha scenario?

Response: Thank you very much for your question. We apologize for not clarifying this point. To be conservative in our estimation, we assumed that a biomass application rate of 20t/ha would achieve the optimal yield increase reported in the literature. In other words, 20t/ha is considered the optimal application rate. We added the description in the Method section - Yield increasing.

However, it is evident that the actual biochar production from biomass conversion is limited on most lands and significantly lower than 20t/ha. Therefore, the actual yield improvement effects of per ton of biomass converted to biochar is low. The formula can be seen in Supplementary Note 4.2.

4. Yield increases:

4.1. Overall, I assume the effect of yield increases to be highly overrated in this assessment. In general, the numbers assumed for yield increases are particularly high, compared to other meta-analyses (i.e. Ye et al. (2020), Jeffery et al (2017)). It is not discussed how these high numbers relate to these former findings. It is unclear which reference was used for the yield increase of each crop. Further, the references used are rather unsuitable. Major et al. (2019) analyse very specific conditions for maize that are not applicable in this study. Wang et al (2019) evaluate the effect of co-composted biochar, which includes a significant effect of the fertilization from the compost. The effect is not separated in the meta-analysis because not all studies include fertilized controls. However, the Chinese agricultural includes vast areas of high fertilizer application rates. These numbers are thus not useful here.

Response:

Thank you for your insightful comments regarding the assessment of yield increases in our study. We greatly appreciate your thorough review and the opportunity to address these concerns. Your input has prompted us to re-evaluate the impact of biochar on crop yield improvement in the context of Chinese farmland.

Upon conducting a detailed comparison between the latest meta-analyses specific to China and global meta-analyses, we have found that the effect of biochar on crop yield improvement in Chinese farmland is notably lower when compared to the global average. This finding suggests that our previous estimation of the benefits associated with increased yield may have been overestimated. To rectify this, we have made adjustments to the yield increase estimates specific to China in Supplementary Note 4.2. By incorporating the new estimate, we have revised the assessment of crop yield benefits in the main text, specifically in Line 184-190 and Figure 2c. These modifications now reflect the more accurate estimation of yield increase for Chinese farmland.

‘Biochar also offers additional application incentives through its ability to bolster crop yields. However, the estimated co-benefits from yield improvements in our study are minor, offsetting only 1% to 22% of costs. This is due to the prevalent high-rate fertilizer application in Chinese fields, which makes the yield enhancement impact of biochar comparatively less significant than suggested by international studies. For instance, the latest meta-analysis indicates that biochar application only improves major crop yields in China by approximately 10%⁴⁶, a figure significantly lower than the global average of 35%²⁹.’

References:

1. Xia, L. et al. Integrated biochar solutions can achieve carbon-neutral staple crop production. *Nat Food* 1–11 (2023) doi:10.1038/s43016-023-00694-0.
2. Singh, H., Northup, B. K., Rice, C. W. & Prasad, P. V. V. Biochar applications influence soil physical and chemical properties, microbial diversity, and crop productivity: a meta-analysis. *Biochar* 4, 8 (2022).

4.2. Also, it is not clear from the description in the SI whether you use average yields for all areas of one crop, i.e. wheat, or whether you use disaggregated values, like the ones you use for the residue calculations.

Response:

Thank you for bringing this issue to our attention. We appreciate your insightful feedback and have carefully addressed your concerns regarding the use of average yields in our analysis.

In the previous version of the manuscript, we indeed utilized average yields for each crop across all grid cells. However, based on your suggestion, we have made significant revisions in the updated version of the paper. We now incorporate disaggregated yield data specific to different crops and grid cells, allowing for a more accurate and nuanced analysis.

To provide clarity on this matter, we have included the description in Supplementary Table 14 and its note, outlining the methodology and data sources used to capture the variation in crop yields across different regions and grid cells.

5. **Structure:**

5.1. Figure 5: all maps need letters for identification, i.e. a), b), c), d)

Response: Thank you for pointing it out. We have added letters for all figures in the revised paper.

5.2. Methods: The methods section is partly lacking relevant information.

1) For example, in the paragraph of forest residues, you mention statistics and gridded data without referencing it. Also, for the coefficients it would be good to know how they are derived or if they are literature-based. The same is true for the residue-to-product ratio of grassland.

Response:

Thank you for bringing these points to our attention. We appreciate your careful review and have taken steps to address the concerns raised in your feedback.

In response to the mention of statistics and gridded data in the paragraph on forest residues, we have added appropriate references to support the information presented. These references can be found in Line 335-344 of the revised paper.

‘To clarify the available forestry residues and their spatial distribution, we started from the statistics on planted areas and production at 2018 level^{58,59}, and adopted method developed by Fu et al. to calculate 10 types of available forestry residues using processing and trimming coefficients of various types of forestry residues⁶⁰. Then, available forestry residues were assigned to grids based on land use type and gridded Net Primary Productivity (NPP) in 2018. The spatial data of wooded and other wooded land were selected from the Resource and Environment Science and Data Center (RESDC) with resolution of $1 \text{ km} \times 1 \text{ km}$ ⁶¹. NPP at the 2018 level was obtained from the Moderate-resolution Imaging Spectroradiometer (MODIS) product-MOD17A3HGF.006-with resolution of $500 \text{ m} \times 500 \text{ m}$ ⁶². The formula used for estimation of forestry residues and data sources can be found in Supplementary Note 1.2.’

Regarding the coefficients used in our method, we made a clear description in Supplementary Table 2 and 3, mainly from Statistical Yearbook and Fu et al.

In terms of grass residues, the modification is reflected in Line 345-348:

‘Initially, we sourced statistical data on available grassland residues from natural grasslands across the country from the 2018 Annual Report on China's Forestry and Grassland Development⁶³. Then, grassland residues was assigned to each grid based on NPP that was obtained from MOD17A3HGF.006⁶² and spatial land use type data of high-, middle-, and low-coverage grassland were accessed from RESDC⁶¹.’

2) In the part on slow pyrolysis, you mention that temperature, duration, and equipment are important, yet give no information on i.e. what temperature you assume. Also, it is not clear, whether you assume the produced syngas to be used for power generation or if another source is used.

Response:

Thank you for raising these important points regarding the slow pyrolysis section of our manuscript. We appreciate your feedback, and we have carefully addressed these concerns in our revised version.

In the revised method section, we have provided more detailed information regarding the temperature assumption for slow pyrolysis. Additionally, we now explicitly state that we assume the syngas is used for industrial steam production based on site survey in typical biochar plants in China.

Please refer to Line 385-393 in the revised manuscript for the updated information on slow pyrolysis, which addresses these points in greater detail.

‘Slow pyrolysis. We assumed the deployment of pyrolysis plants, producing both biochar and syngas, at the center of each grid. To standardize the output from varying types of biomass, we adopted a unified accounting framework by Woolf et al.³², with the empirical foundation rooted in the existing literature. We used the physicochemical properties of various biomass types as inputs, and sets the pyrolysis temperature at 550°C for this study. Further, we adopted the biomass gasification technology that co-produces biochar and syngas, as seen in pilot projects and experimental literature. Biochar's heating value was calculated based on the empirical analyses conducted by Qian et al⁶⁷. Syngas production was aligned with energy balance and was subsequently used to generate industrial steam. For more in-depth information, refer to Supplementary Note 2.’

3) In the section on negative emission potential, you need to at least address which approach or database you follow for the biochar yield and carbon content in the char – this information is crucial for the overall NE potential and currently not described sufficiently. Also, you need to clarify the period of time here.

Response: Based on the revised method, we revised the negative emission potential part, as shown in line 394-400 of the revised paper:

‘The value was determined based on the quantity of available biomass feedstocks, the weight conversion rate from feedstock to biochar, the carbon content of biochar, and the decomposition rate of biochar during 100 years, which were calculated based on accounting framework developed by Woolf et al.³² and chemical composition of crops planted in China. Relevant formulas and data can be found in Supplementary Note 2&3.’

4) Also, you need to describe how you quantify the yield increases in the Methods section.

Response:

Thank you for addressing this concern. To provide a clearer understanding of how we quantify the yield increase, we have included an additional paragraph in the method section of the revised manuscript.

This paragraph, located in Line 401-407, provides a detailed description of the methodology employed to quantify the yield increase. By incorporating this information, we aim to enhance the transparency and reproducibility of our research, allowing readers to better comprehend the approach used to measure the observed changes in yield.

‘Yield increasing. We calculated the benefits of optimal biochar application rate-induced yield improvement by multiplying gridded crop yield⁵⁷, crop prices, and the rate of yield increase⁴⁶. Subsequently, we computed the benefits of the actual biochar application rate, which was determined by the crop yields and the weight conversion

rate from biomass to biochar, using the ratio of the actual to the optimal application rate. For conservative estimation purposes, we set the optimal biochar application rate at 20 t ha⁻¹²⁹. Relevant formulas and data can be found in Supplementary Note 4.2.’

5.3. Comments on understanding:

1) p.8 : “For example, the share of agroforestry residue is 80% in East China and 66% in Central and South China, with low average costs of 66.7 and 68.1 \$/tCO₂, respectively (Fig. 5c).”

- What is the 80% referring to? Where do I find 5c?

Response:

Thank you for bringing this issue to our attention. We appreciate your careful review and have noted the correction. The reference to "80%" in our previous version should actually pertain to the share of agroforestry residue within the overall biomass feedstocks. We apologize for the error.

To rectify this, we have revised the specific line in question, now reflected as "4d" instead of "5c," in order to accurately represent the intended information. The revision has been made in Line 227-229 of the revised manuscript.

‘For example, the share of agroforestry residue in total available biomass feedstocks is 85% in East China and 70% in Central and South China, with low average costs of 64.0 and 69.2 \$ t⁻¹CO₂, respectively (Fig. 4d).’

2) p.9: “To date, no biochar offset credit projects have been implemented, mainly because of the lack of suitable baselines and monitoring methodologies.”

- What about the voluntary market?

Response:

Thank you for bringing this matter to our attention. Upon further investigation into policy practices, we have identified the existence of voluntary markets that have established platforms specifically related to carbon removals and biochar. This important information has prompted us to revise our paper accordingly.

To reflect the progress in policy and the presence of these platforms, we have made revisions in Line 267-270 of the manuscript. These revisions now incorporate the relevant details and acknowledge the advancements in this area.

‘Yet, promising initiatives have been seen in the United States, Finland, and beyond, where organizations have established voluntary carbon removal platforms that incorporate biochar and have started to explore validation and monitoring methodologies⁴⁸’

3) p.10: “Additionally, further strengths and drawbacks of biochar implementation should be examined carefully to minimize the trade-offs with key sustainable development goals, e.g., heavy metal sorption, loss of biodiversity⁴³, and land use change emissions.”

- Loss of biodiversity is a trade-off, while heavy metal sorption is a co-benefit. Also, it needs to be clarified with what these trade-offs are associated, i.e. feedstock generation or biochar application to soils.

Response: Thank you for pointing out this issue. We revised our paper accordingly in line 304-308.

‘Lastly, our analysis does not fully address the environmental and socio-economic trade-offs associated with the large-scale deployment of biochar-NETs. Increasing biomass demand could potentially lead to emissions from land-use changes, heightened competition with food production, and biodiversity reduction⁵². Simultaneously, the application of biochar can also offer additional co-benefits, such as enhanced heavy metal adsorption in soils²¹.’

Reviewer #2 (Remarks to the Author):

The manuscript describes the potential negative emission potential, economics and spatial deployment of biochar applications in China. Although the topic is within the scope of the journal, the current version of the manuscript does not present a clear novelty compared to other papers on the field. Please, find some suggestions for improvement below:

Dear reviewer, thank you for your insightful comments. We sincerely appreciate your recognition of the topic addressed in our paper. Your feedback has helped us identify the shortcomings and areas for improvement, and we have made significant revisions accordingly. Allow us to outline the key improvements made in this revised version and address your concerns point by point.

1. Research questions could be better elucidated. Apart from the "limited to research on agricultural residues", are there limitations in the current assessment methods or data?

Response:

Thank you for bringing up this important point. We appreciate your feedback, and we have carefully revised our paper to address the limitations in existing studies. In our revised article, we have provided a more comprehensive discussion on the constraints related to data availability and assessment methods.

To summarize, we have identified three major limitations in previous studies. Firstly, the focus of previous research has primarily been on agricultural residues, while neglecting the potential contributions from other sources such as forests, grasslands,

and energy crops. This narrow focus has limited our understanding of the complete biochar potential within different feedstock categories.

Secondly, previous studies have failed to recognize the significant disparities that exist in the properties of biochar derived from different feedstocks. This lack of recognition has hindered the development of a robust basis for spatially explicit estimates of biochar potential. By not accounting for these variations, previous assessments may have provided incomplete or inaccurate evaluations of biochar potential.

Lastly, an important aspect that has been overlooked in previous studies is the consideration of crop yield improvements resulting from variations in soil conditions. Soil properties, such as texture, pH levels, and biochar application rates, can greatly influence the efficacy of biochar in enhancing crop productivity. By disregarding these factors, previous assessments may have underestimated the overall potential of biochar.

To address these limitations, we have made significant changes in Line 59 to 77 of our revised article. These modifications are listed as follows:

‘To date, the feasibility of biochar deployment in China has been a subject of extensive study, with a primary focus on the near-term aggregated potential derived from agricultural residues^{26,27}. However, previous investigations have often overlooked crucial factors such as the diverse range of feedstocks, the varied physicochemical properties of biochar²⁸, and the disparities in soil conditions²⁹. These omissions significantly impact the evaluation of the negative emissions potential and economic costs associated with biochar deployment. Firstly, previous estimates primarily concentrated on agricultural residues, neglecting the substantial contributions from forestry residues, grassland residues, and potential energy crops, therefore leading to a systematic underestimation of the overall biochar potential within the country. Additionally, much of existing studies have inadequately captured the physicochemical variations in biochar derived from different crops, which hinders the assessment of biochar economics and negative emissions on a spatial scale. Furthermore, the disparities in soil conditions, including soil texture, pH levels, and biochar application rates, have not been adequately incorporated into the assessment of the benefits derived from enhanced crop yields^{29,30}. The omission of these crucial factors further complicates the prioritization of locations and feedstocks for biochar deployment. However, recent years have witnessed substantial advancements in the field, particularly with the increased availability of data resulting from field experimental results³¹. These advancements, coupled with advanced methodologies³² and improved granularity of spatial data³³, now enable a more comprehensive and spatially explicit assessment of the negative emission potential and economics of biochar.’

2. Lines 59-61 "In fact, the negative emission potential and cost of biochar derived from various feedstocks differ greatly owing to their disparate properties, namely, porosity, surface area, and carbon content." The chapter on cost of negative emissions

from biochar only focuses on carbon content. How do the other two factors affect economics without any discussion?

Response:

Thank you for your valuable feedback, and we appreciate your attention to the specific statement in question. We apologize for any confusion caused by our unclear wording.

To address this concern, we would like to explain the intended meaning behind the statement. The higher porosity and surface area of biochar play a significant role in reducing nutrient loss and enhancing crop yield. These effects can have a substantial impact on the cost-benefit analysis and the overall economic viability of biochar. While we did not explicitly discuss the direct influence of porosity and surface area on economics, their effects are implicitly embedded in the yield improvement effects, which are critical factors in cost-benefit assessments.

To avoid any further confusion, we have decided to remove the sentence in question during the revision. By doing so, we aim to provide a clearer and more concise presentation of our research findings without misleading or incomplete statements.

3. For the cost analysis of negative emissions of biochar, please include cost breakdown and financial assessment, also suggest adding the storage cost of biomass.

Response:

Thank you for your valuable advice. We appreciate your suggestions, and we have implemented them to enhance the economic analysis framework. In line with your recommendations, we have employed a standardized cost breakdown and financial assessment approach to evaluate the economics of negative emissions achieved through biochar.

Specifically, we have adopted the commonly used cost-benefit analysis (CBA) method in financial analysis to calculate the unit net cost of negative emissions (UC, \$/tCO₂). This calculation takes into account the net present value (NPV) and the total negative emissions over the project's lifetime (TNE). This approach provides a comprehensive assessment of the economic viability of biochar in terms of its contribution to negative emissions.

Furthermore, in response to your suggestion and in accordance with our survey data, we have included the storage cost of biomass in our analysis. This addition allows for a more accurate and comprehensive evaluation of the overall costs associated with biochar production and storage.

To ensure clarity and consistency, we have made revisions to Figure 2 to incorporate the cost breakdown, providing a visual representation of the different cost components. Additionally, we have updated the Economic Analysis section in the Methods, as reflected in Line 408-422 of the revised paper.

‘Economic analysis. Cost-benefit analysis was adopted to analyze the economics of biochar, which is one of the commonly used financial assessment method to evaluate the project value by comparing the costs and benefits. Here, the system boundaries were defined as feedstock purchasing, transportation, storage, pyrolysis, biochar transport, application, and effectiveness on crops. It was assumed that feedstock would be transported to a pyrolysis plant located in the center of each grid, while biochar would be returned to the fields in which the feedstock was harvested. First, net present value (NPV) of 20-year project of pyrolysis plant was calculated. NPV includes the initial investment, annual cash inflows and outflows. Annual cash inflows consisted of increasing yield income and syngas income calculated based on the price and production. Annual cash outflows consisted of the cost of feedstock purchasing, feedstock storage, operation and maintenance, transport of biomass and biochar, and biochar application. Then, the cost of negative emission can be defined as the initial investment and cash outflows apportioned to each unit of CO₂ captured in biochar during the whole period. The net cost of negative emission can be considered as the opposite of the NPV apportioned to each unit of CO₂. Relevant formulas and data can be found in Supplementary Note 5.’

4. In the chapter on spatial analysis of biochar potential, only soil texture is mentioned as an influencing factor on crop yield, soil pH is also a key factor on crop yield and is suggested to be added.

Response:

Thank you for your valuable suggestion. We appreciate your attention to the influence of soil pH on crop yield, and we agree that it is an essential factor to consider. In our analysis, we have incorporated the pH value as a component of soil acidity, which directly affects yield improvement. To ensure clarity and completeness, we have revised the section on spatial explicit analysis in the Methods section, specifically in Line 424-428, to explicitly address the inclusion of soil PH level.

‘In addition to clarifying the patterns of feedstock distribution, soil texture, PH and biomass abundance in different regions were considered. First, data of soil texture and PH level were adopted from the Harmonized World Soil Database v1.2⁶⁶. With reference to Singh et al.²⁹ and Wang et al.⁴⁷, biochar applied to soil with either coarse and fine texture or acidity was assumed more effective in improving crop yields, as shown in Supplementary Note 4.2.’

We also revised Supplementary Note 4.2 as follows:

‘Given that biochar applied to soil with coarse or fine texture has been found to be more effective in improving crop yields, and soil pH also has an influence on the effectiveness of biochar application, we assumed that the effects of yield increase on coarse- and fine-texture soil or acidic soil would be 50% greater than the average value, while that on medium-texture soil or alkaline soil would be only half the average value with reference to Singh et al.’

5. The paper mentions that biochar has negative emission potential, which is consistent with China's goal of achieving carbon neutrality by 2060. Could a dynamic assessment model be used to simulate the negative emission potential of biochar from now until 2060?

Response:

Thank you for your valuable suggestions. The dynamic assessment of the negative emission potential of biochar is indeed an important research topic that warrants attention. However, our literature survey indicates that the complexity of this issue extends beyond the scope of our current analysis. This complexity arises due to the interactions among climate change scenarios, socio-economic scenarios, and the assumptions surrounding mitigation and adaptation policies (please refer to Table 1 and Table 2 in the following response). As a result, conducting a comprehensive dynamic assessment that encompasses the period until 2060 would require a more extensive and specialized investigation.

In light of these considerations, we have chosen to adopt a Monte Carlo analysis approach. By incorporating uncertainty ranges of key parameters that reflect dynamic scenario uncertainties, we aim to ensure the robustness of our results to the greatest extent possible. This approach allows us to capture a range of potential outcomes and provide valuable insights into the potential negative emission impact of biochar.

We have made revisions to the main results section, specifically in Line 154-158, as well as in Supplementary Note 7, to reflect the outcomes of the Monte Carlo analysis. These changes enhance the clarity and accuracy of our findings, providing a more robust assessment of the negative emission potential of biochar.

'We also conducted a sensitivity analysis to show the long-term impact of climate change on the estimation of biochar potential. Without additional consideration for land use pattern, the Monte Carlo analysis suggest a slight growth of the negative emission potential of biochar under all scenarios. This growth is primarily attributable to the rise in forestry residues, which are affected by climate change expected in the second half of the 21st century⁴²⁻⁴⁴ (Supplementary Note 7).'

We conducted a literature review and consulted relevant experts and concluded that a comprehensive dynamic assessment that encompasses the period until 2060 would require a more extensive and specialized investigation due to following reasons.

Firstly, for agricultural residues, their calculation is based on crop production. To estimate future crop production, a common approach is to quantify the climate impact on crop yield and incorporate it into crop models or economic models to obtain crop production projections under various scenarios. Through our literature review on major grain crops, as shown in Table 1, we found that there is significant uncertainty in yield changes for major grain crops in the second half of the 21st century. For example, the

increasing rate of the rice yield could range from -9% to 5.6%, wheat from -21% to 13%, and maize from -29% to 41% [1-6]. Furthermore, under socio-economic scenarios, the growing demand for food may lead to an estimated dependence on imports of agricultural products ranging from 15% to 24% by 2050 [7]. This implies that domestic production is influenced by not only domestic demand.

Table 1 Changes of crop yield and production under various scenarios in existing studies

Study	Methods	Crop types	Scenarios			
Xie et al., 2018 ¹	CAPSiM-GTAP	Crop production	RCP 2.6		RCP8.5	
			2030	2050	2030	2050
		Rice	-0.28%	-0.55%	-0.21%	-0.22%
		Wheat	-0.97%	-2.21%	-1.92%	-4.03%
		Maize	0.4%	3.58%	1.01%	1.93%
		Crop yield	RCP 2.6		RCP8.5	
			2030	2050	2030	2050
		Rice	-0.56%	-1.34%	-0.78%	-2.6%
Wheat	-2.28%	-4.83%	-3.39%	-9.39%		
Maize	0.33%	0.25%	-0.01%	0.31%		
Cui et al., 2022 ²	CAPSiM	Crop yield (%)	RCP 2.6		RCP8.5	
			-0.59	-1.51	-0.75	-2.62
		Rice	(-0.61~ -0.56)	(-1.60~ -1.41)	(-0.80~ -0.71)	(-2.87~ -2.38)
		Wheat	-2.62	-5.79	-3.68	-10.63
		Maize	0.28	0.10	-0.09	0.13
Wang et al., 2021 ³	ClimaCrop-GTAP	Crop yield	<1.5°C ~4°C	Crop production	<1.5°C ~4°C	
		Rice	0.4%-5.6%	Rice	0.23-0.49 Mt	
		Wheat	-2%~-12.5%	Wheat	-0.3 Mt to 0 Mt	
Liu et al., 2016 ⁴	grid-based point-based statistical regressions	Crop yield	1 °C global temperature increase			
		Wheat	-5.9% ~ 2.5%			
			-5.3% ~ -1.8%			
Zhang et al., 2017 ⁵	CLIMATE MODELS: PCM、	Crop yield change in log	2070-2099			
			B1	A1B	A2	

	ECHAM、 CGCM、 CCSM、 HadCM3	Rice	-2 ~ 0.3	-4 ~ -0.4	-5 ~ -0.5	
		Wheat	-3 ~ 0.3	-6 ~ -0.4	-5 ~ -0.9	
		Maize	-3 ~ 0.8	-6 ~ -0.3	-7 ~ -0.3	
Zhao et al.,2017 ⁶	global grid-based and local point-based models, statistical regressions, and field-warming experiments	Crop yield	1 °C global temperature increase			
		Rice	-9% ~ 5%			
		Wheat	-16% ~ 13%			
		Maize	-29% ~ 9%			

Furthermore, the calculation of forestry residues is also based on the forestry production. Factors such as temperature rise, CO₂ fertilization effect, forest management, and socio-economic scenarios all influence woody biomass production. Studies have shown that under current climate conditions, carbon stocks are expected to significantly increase by 2060, and may even increase further under climate change scenarios like RCP4.5 and RCP8.5 [8, 9], as shown in Table 2 below. Assuming linear growth, the increasing rate of biomass stock could range from 11% to 192% by 2060, with the median being 91%. Compared to climate and CO₂ concentration changes, forest management practices have a more significant impact on the stem biomass accumulation in forests [10]. Additionally, under different shared socio-economic pathways, harvest volumes can vary by a factor of two or even more [11].

Table 2 Changes of forest biomass under various scenarios in existing studies

Study	Methods	Scenarios		2001-2100		
		Scenarios	C storage (tC ha-1)	NPP (t ha-1 yr-1)		
Jin et al., 2022 ⁸	TRIPLE X 1.6	Current	2014: 37.12 ± 28.56	2014: 3.29 ± 1.29		
			2030: 65.74 ± 31.21	2030: 3.54 ± 0.97		
			2060: 113.71 ± 38.70	2060: 3.54 ± 0.69		
		RCP4.5	2030: 69.26 ± 33.40	2030: 3.69 ± 1.05		
			2060: 119.27 ± 41.62	2060: 3.67 ± 0.73		
RCP8.5	2030: 69.65 ± 33.67	2030: 3.70 ± 1.05				
		2060: 119.87 ± 41.97	2060: 3.68 ± 0.74			
Dai et al., 2016 ⁹	LANDIS-II and PnET-II	Scenarios	Aboveground biomass (AGB) change (g m ⁻²)	Compared with the control scenario in 2100		
			2010-2100			
			Current		1540.7	
			RCP2.6		1911.8	24.10%
			RCP4.5		2529.9	64.20%
RCP8.5	1999.5	29.80%				

Zhou et al., 2013 ¹⁰	InTEC - G4M	Current climate and CO2 concentration	Maximum average increment	increasing before 2050, then decline
			Maximum average biomass	1.9 kg C/m ² - 7.6 kg C/m ²
		Changing climate and current CO2 concentration	Maximum average increment	1.9 kg C/m ² - 11.2 kg C/m ²
			Maximum average biomass	Fluctuates within 1.9-3.1 kg C/m ²
		Changing climate and CO2 concentration	Maximum average increment	6.0 kg C/m ² 至 25 kg C/m ²
			Maximum average biomass	2.0 kg C/m ² to 6.0 kg C/m ²

In light of previous consideration, we have chosen to adopt a Monte Carlo analysis approach by incorporating uncertainty ranges of key parameters (Table 3) that reflect dynamic scenario uncertainties.

Table 3 Ranges of yield changing from existing studies

Biomass types	low	median	high
Rice	-9%	-0.77%	5.6%
Wheat	-21%	-4.11%	13%
Maize	-29%	0.28%	41%
Forestry residues	11%	91%	192%
Energy crops	-3.30%	-0.60%	13%

References:

1. Xie, W. et al. Climate change impacts on China's agriculture: The responses from market and trade. *China Economic Review* 62, 101256 (2020).
2. Cui, Q., Ali, T., Xie, W., Huang, J. & Wang, J. The uncertainty of climate change impacts on China's agricultural economy based on an integrated assessment approach. *Mitig Adapt Strateg Glob Change* 27, 25 (2022).
3. Wang, D. et al. Economic impacts of climate-induced crop yield changes: evidence from agri-food industries in six countries. *Climatic Change* 166, 30 (2021).
4. Liu, B. et al. Similar estimates of temperature impacts on global wheat yield by three independent methods. *Nature Clim Change* 6, 1130–1136 (2016).
5. Zhang, P., Zhang, J. & Chen, M. Economic impacts of climate change on agriculture: The importance of additional climatic variables other than temperature and precipitation. *Journal of Environmental Economics and Management* 83, 8–31 (2017).
6. Zhao, C. et al. Temperature increase reduces global yields of major crops in four independent estimates. *Proceedings of the National Academy of Sciences* 114, 9326–9331 (2017).

7. Zhao, H. et al. China's future food demand and its implications for trade and environment. *Nat Sustain* 4, 1042–1051 (2021).
8. Zhou, L. et al. Carbon dynamics in woody biomass of forest ecosystem in China with forest management practices under future climate change and rising CO₂ concentration. *Chin. Geogr. Sci.* 23, 519–536 (2013).
9. Jin, J. et al. Stand carbon storage and net primary production in China's subtropical secondary forests are predicted to increase by 2060. *Carbon Balance Manage* 17, 6 (2022).
10. Dai, E., Wu, Z., Ge, Q., Xi, W. & Wang, X. Predicting the responses of forest distribution and aboveground biomass to climate change under RCP scenarios in southern China. *Global Change Biology* 22, 3642–3661 (2016).
11. Lauri, P. et al. Global Woody Biomass Harvest Volumes and Forest Area Use Under Different SSP-RCP Scenarios. *JFE* 34, 285–309 (2019).

6. What are the limitations of the proposed analysis? Close out in either the methodology or discussion. Make sure to discuss how this would affect the results; then highlight again in the conclusion and finger towards further research you or other could undertake to improve.

Response:

Thank you for raising this important issue. We have carefully addressed your comment and made revisions to address the limitations of our proposed analysis. Specifically, we have included a dedicated section in the discussion that outlines the limitations and their potential impact on the results.

In this section, we emphasize the limitations related to other feedstocks with proven capacity, dynamic factors such as technology progress, and the consideration of trade-offs and co-benefits. By acknowledging these limitations, we provide a more comprehensive understanding of the scope and applicability of our analysis.

Furthermore, in the conclusion section, we have reiterated the discussion on limitations and their potential influence on the results. We have also provided suggestions for further research that can be undertaken to improve upon these limitations.

To access the revised text, please refer to Line 295-312 in the revised manuscript.

‘Our study has some limitations. First, our analysis does not encompass all types of biomass resources. We focused on specific feedstocks, leaving out others like livestock manures⁵⁰, which have proven potential for biochar production. Moreover, the biochar co-production technology we employed, while economically viable, does not prioritize biochar yield maximization. As carbon budgets become more restrictive, the balance between negative emissions and energy value in biochar production⁵¹ warrants careful consideration. These elements might lead to a potential underestimation of negative emission capacities in our study. Second, our research does not fully account for dynamic influences such as technological advancements, economies of scale, and evolving carbon prices, which could all affect biochar's economic viability and

potentially lower its future cost. Lastly, our analysis does not fully address the environmental and socio-economic trade-offs associated with the large-scale deployment of biochar-NETs. Increasing biomass demand could potentially lead to emissions from land-use changes, heightened competition with food production, and biodiversity reduction⁵². Simultaneously, the application of biochar can also offer additional co-benefits, such as enhanced heavy metal adsorption in soils²¹. Despite these limitations, our key finding remains that biochar is a promising NET option for achieving carbon neutrality goals in China and should be included in the climate change mitigation toolbox. Future research should aim to explore these potential trade-offs and co-benefits, offering a more holistic understanding of biochar's role in climate change mitigation⁵³⁻⁵⁵.

7. What are the potential sources of error in this analysis, and how can the underlying assumptions be improved?

Response:

In the revised paper, we highlighted the potential uncertainty source in Line 277-286. We also conducted a Monte Carlo analysis incorporating those uncertainty and highlight the most sensitive parameter which can be further improved in future research.

‘While this study employs a robust framework and incorporates data from the latest experimental literature and pilot projects, the potential of biochar remains subject to considerable uncertainty. This uncertainty primarily originates from several factors: availability of biomass resources, properties of various biochar types, pyrolysis techniques and conditions, and the impact of biochar application on crop yield. To evaluate this uncertainty, we conducted a Monte Carlo analysis (see Supplementary Note 7). The mean estimate of biochar's negative emission potential was 0.86 GtCO₂ yr⁻¹ under the sustainable technical scenarios, ranging from 0.55 to 1.17 GtCO₂ yr⁻¹. The mean negative emission cost of biochar was \$88 t⁻¹CO₂, with a range from -\$42 to \$219 t⁻¹CO₂. Of all the parameters, those related to by-product income and feedstock purchasing costs were the most sensitive, with the potential to alter biochar cost by over 100% (see Supplementary Figs. 12&13).’

REVIEWER COMMENTS

Reviewer #1 (Remarks to the Author):

I commend the authors for the remarkable enhancements made to the manuscript. It is evident that my comments were thoughtfully considered, and a significant amount of effort was invested in refining the methodology and research design. I anticipate that the paper will substantially benefit from the added scenario scope in the revisions. The resulting figures and contextualization have harmonized well with other internationally recognized work in this field.

The presentation of the study design and potentials in Table S8 is particularly well-organized and clear. In my view, incorporating this valuable overview into the main body of the article could enhance its accessibility. If it aligns with the journal's formatting guidelines, I suggest considering the inclusion of this overview in the main body.

Beyond this, I have no further points of critique to offer and confidently recommend the publication of the revised manuscript.

Reviewer #2 (Remarks to the Author):

This manuscript has certain novelty. However, there are some comments which need to be addressed to improve the quality of the paper.

1.This paper assumes that biochar is 75% permanence rate over a 100-year period. Is biochar applied once or annually? If applied annually, what is the number of years of continuous application? Is the half-life of biochar-carbon the same for different biomass production?

2.The period of biochar application will also affect the negative carbon potential of biochar. Is biochar specified in the paper to be applied at a specific time period?

3.In estimating the long-term impact of climate change on the negative carbon potential of biochar, whether the scenarios include a combination of Shared Socioeconomic Pathways (SSPs) and Representative Concentration Pathways (RCPs).

4. How is the yield of syngas in by-products calculated? Syngas contains CH₄. When estimating the negative carbon potential of biochar, is the reduction of CH₄ emissions from the recovery of syngas taken into account?

5. The pyrolysis technology and energy consumption used in biochar production have an impact on cost-effectiveness and negative carbon potential. What pyrolysis technology is used? How much is the energy consumption?

Point-by-point Response to Review Comments on Biochar

Reviewer #1 (Remarks to the Author):

I commend the authors for the remarkable enhancements made to the manuscript. It is evident that my comments were thoughtfully considered, and a significant amount of effort was invested in refining the methodology and research design. I anticipate that the paper will substantially benefit from the added scenario scope in the revisions. The resulting figures and contextualization have harmonized well with other internationally recognized work in this field. The presentation of the study design and potentials in Table S8 is particularly well-organized and clear. In my view, incorporating this valuable overview into the main body of the article could enhance its accessibility. If it aligns with the journal's formatting guidelines, I suggest considering the inclusion of this overview in the main body. Beyond this, I have no further points of critique to offer and confidently recommend the publication of the revised manuscript.

Response:

Thank you very much for the comments you provided previously, which was highly valuable. We appreciate your recognition of the efforts we have put into the revised manuscript. We have made the following changes to the new version:

Firstly, we have moved *Table S8 Potentials of available biomass feedstocks under various scenarios* to the main body of the manuscript as Table 1, as you recommended.

Secondly, when we check *Table S6 Carbon contents and weight conversion rates of biochar produced from various feedstocks* in this round of revision, we noticed an unrecognized mistake related to weight conversion rate in our last revision. We made a mistake to use the weight conversion rate and carbon content in different concept scopes: ash-free biochar, and biochar with ash.

Due to the importance of these two parameters, we believe it is important to take this opportunity to take this issue back to your attention and explain our revision in this round accordingly.

To be specific, weight conversion rate and carbon content are two key factors to calculate negative emission potential of biochar. Using empirical framework developed by Woolf et al.¹, we got the weight conversion rate (α_i^{DAF}) and carbon content (β_i^{DAF}) of dry ash-free (DAF) biochar. Considering ash content, we then calculated the weight conversion rate (α_i) and carbon content (β_i) of the resultant biochar (with ash). When calculating negative emissions potential (Seq_i), we can use either Formula 1 or Formula 2, and the results from both formulas are identical.

$$Seq_i = \alpha_i \cdot \beta_i \cdot \frac{44}{12} \cdot \text{permanence rate} \quad (1)$$

$$Seq_i = \alpha_i^{DAF} \cdot \beta_i^{DAF} \cdot \frac{44}{12} \cdot \text{permanence rate} \quad (2)$$

However, we mistakenly used α_i^{DAF} and β_i to calculate the negative emission potential. α_i is higher than α_i^{DAF} since the former includes the ash content from the biomass (Bm_ash_i). Therefore, we underestimate the negative emission potential in the earlier version.

In this round of revision, using the approach developed by Woolf et al.¹, we calculated weight conversion rates from rice straw to DAF biochar as 22.8% (α_i^{DAF}), which plus the rice-straw ash content of 13.4%, resulting in a total of 36.4%. In other words, the weight conversion rates from rice straw to resultant biochar is 36.4% (α_i). On contrary, α_i^{DAF} is 22.9% for maize cob, while its ash content is merely 3.8%, resulting in α_i of 26.8%. We made the revision in SI Note2. The formula and data of various types of biochar also can be seen as follows:

$$\beta_i = \beta_i^{DAF} * (1 - Bc_ash_i) \quad (2-1)$$

$$\beta_i^{DAF} = 0.93 - 0.92 * e^{-0.0042*T} \quad (2-2)$$

$$Bc_ash_i = Bm_ash_i / \alpha_i \quad (2-3)$$

$$\alpha_i = \alpha_i^{DAF} + Bm_ash_i \quad (2-4)$$

$$\alpha_i^{DAF} = 0.1261 + 0.5391 * e^{-0.004*T} + 0.2733 * L_i \quad (2-5)$$

$$Seq_i = \alpha_i \cdot \beta_i \cdot \frac{44}{12} \cdot \text{per} \quad (3-1)$$

Table S6 Carbon contents and weight conversion rates of biochar produced from various feedstocks.

Data	Rice		Maize			Other			Energy crops	
	Straw	Husk	Wheat straw	Straw	Cob	agricultural residues	Grass residues	Forestry residues	miscanthus	Sweet sorghum
α_i^{DAF}	22.8%	25.2%	21.5%	22.3%	22.9%	23.4%	21.8%	25.3%	21.9%	24.2%
α_i	36.2%	42.9%	28.6%	28.6%	26.8%	29.2%	28.8%	27.5%	24.6%	25.0%
β_i	52.9%	49.3%	63.0%	65.2%	71.9%	67.2%	63.5%	77.2%	74.7%	81.0%
Bm_ash_i	13.4%	17.7%	7.1%	6.4%	3.8%	5.8%	7.0%	2.2%	2.7%	0.9%
L_i	15.6%	24.3%	10.7%	13.5%	15.9%	17.7%	11.8%	24.7%	12.3%	20.4%

The calculation using the empirical framework harmonizes with experimental findings well, in terms of both dry ash-free biochar and resultant biochar with ash. In the existing experiments, both of the weight conversion rate from biomass to dry ash-free (DAF) biochar and the resultant biochar (with ash) are frequently demonstrated for different research purposes. Experiments that make comparison of biochar produced from different biomass feedstocks usually find rice-based biochar has higher yields than others, mainly due to its significantly higher ash content compared to others. For instance, Wang et al.² observed significant variations in the biochar yields of seven types of feedstocks. After a 4-hour reaction at 500°C, rice husks and straw exhibited

weight conversion rates of 37.6%, while bamboo and maize stalks showed rates of 25.3% and 32.3%, respectively. Even after a 16-hour residence time, rice husks maintained the highest conversion rate at 33.8%, whereas bamboo and maize stalks reached rates of 25.6% and 27.3%, respectively. In terms of rice husk biochar, ash content accounted for more than one-third of the mass. Conz et al.³ also found that, at 550°C, rice husks had a conversion rate of 46.5%, surpassing sugarcane straw (34.6%) and wood chips (36.4%). He et al.⁴ discovered that at 550°C, rice straw had a conversion rate of 36.64%, higher than wheat (33.5%) and corn (35.2%). Regarding dry ash-free biochar, rice straw achieved a weight conversion rate of 28.55%, while wheat and corn were approximately 29.5%.

The consideration of ash in biochar is essential in our study not only for correcting the estimates of negative emissions, but also for reasonable calculation of economics. In practice, biochar transport and application account for biochar that includes ash contents. Ash content also affects the heating value calculation of biochar. Consequently, the new estimates of both negative emission potential and costs in the second round of revision are higher than those in last version. We deeply regret any confusion caused and appreciate your understanding as we rectify this issue.

References:

1. Woolf, D., Lehmann, J., Ogle, S., Kishimoto-Mo, A. W., McConkey, B., & Baldock, J. (2021). Greenhouse Gas Inventory Model for Biochar Additions to Soil. *Environmental Science & Technology*, 55(21), 14795–14805. <https://doi.org/10.1021/acs.est.1c02425>
2. Wang, Y., Hu, Y., Zhao, X., Wang, S., & Xing, G. (2013). Comparisons of Biochar Properties from Wood Material and Crop Residues at Different Temperatures and Residence Times. *Energy & Fuels*, 27(10), 5890–5899. <https://doi.org/10.1021/ef400972z>
3. Conz, R. F., Abbruzzini, T. F., Andrade, C. A. de, Milori, D. M. B. P., & Cerri, C. E. P. (2017). Effect of Pyrolysis Temperature and Feedstock Type on Agricultural Properties and Stability of Biochars. *Agricultural Sciences*, 8(9), Article 9. <https://doi.org/10.4236/as.2017.89067>
4. He, X., Liu, Z., Niu, W., Yang, L., Zhou, T., Qin, D., Niu, Z., & Yuan, Q. (2018). Effects of pyrolysis temperature on the physicochemical properties of gas and biochar obtained from pyrolysis of crop residues. *Energy*, 143, 746–756. <https://doi.org/10.1016/j.energy.2017.11.062>

Reviewer #2 (Remarks to the Author):

This manuscript has certain novelty. However, there are some comments which need to be addressed to improve the quality of the paper.

Response:

Thank you very much for your recognition of the innovation in our revised manuscript. In the following response, we address each of your concerns one by one and strive to improve the quality of the paper. In order to provide a clearer explanation, we have swapped the order of comment 4 and 5 to address the technical aspects and energy consumption first, and then the syngas calculation.

1.This paper assumes that biochar is 75% permanence rate over a 100-year period. Is biochar applied once or annually? If applied annually, what is the number of years of continuous application? Is the half-life of biochar-carbon the same for different biomass production?

Response:

We appreciate your valuable comments and would like to provide further clarification. Given that this study aims at investigating the upper limits of biochar's negative emission potential in one year, our calculations are based on the upper limits of available biomass resources under different scenarios (the current technical scenario, the sustainable technical scenario and the maximum theoretical scenario). In other words, we only care about the potential from supply-side, instead of the actual application time from the dynamic perspective.

$$Seq_i = \alpha_i \cdot \beta_i \cdot \frac{44}{12} \cdot per$$

$$NE = \sum_i Seq_i \cdot Res_i$$

To be specific, the negative emission potential of biochar (NE , tCO₂) was calculated based on Seq_i and Res_i . Res_i implies upper limits of available feedstock biomass i in one year, which includes various agricultural residues, grass residues, forestry residues, and potential energy crops. CO₂ fixed in one unit of feedstock from the atmosphere by photosynthesis, then transferred and “permanently” (over 100 years in our study following definition in literature) preserved in biochar (Seq_i , tCO₂/t biomass) can be calculated using the coefficients of α_i - weight conversion rate from feedstock i to biochar (%), β_i - carbon content of biochar produced by feedstock i (%), and per - the permanence rate of biochar over 100 years (%). The relevant formulae can be seen in SI Note 3.

Certainly, biochar can be applied annually, since the crop residues, forest residues, grass residues and energy crops can be harvested annually. No matter at which year the biochar is produced and applied, carbon from biomass can be fixed in biochar and decompose in certain quantities over the next hundred years. Therefore, we present the results in unit of CO₂·yr⁻¹.

The specific timing of biochar application is contingent upon a range of factors, including the potential of biochar, economic considerations, climate policies, and the need for coordination with other negative emissions technologies. It is indeed a complex issue that would benefit from in-depth dynamic research, for example estimation by Intergrade assessment models (IAMs). Our study is a static study of economics and potential, which aims to establish a foundational understanding on biochar, and provide basic technical and spatial information for IAMs and biochar application in the future.

We apologize for not providing clear information earlier and have emphasized the corresponding text in lines 104-109:

‘Therefore, the ‘Current Technical’ and ‘Sustainable Technical’ scenarios present near-term and mid-term upper limits of biochar’s negative emission potential without compromising food security or habitats. However, it’s crucial to note that both scenarios necessitate progressive policy action to enhance biomass availability beyond current practices. The range of negative emission potentials reported in this paper pertains solely to the scenarios considered and does not reflect constraints imposed by real-world policies.’

Besides, we appreciate your attention to half-life of biochar-carbon. The International Biochar Initiative (IBI) recommends using Hydrogen to organic Carbon molar ratio (H/C_{org}) to determine permanence, which indicates the degree of material condensation¹. As temperature increases, the relative abundance of hydrogen compared to carbon tends to decrease. A lower H/C_{org} corresponds to higher permanence. Biochar produced from different biomass feedstocks and during different pyrolysis conditions varies in H/C_{org} and permanence. However, in cases where H/C_{org} measurements are not feasible, the pyrolysis temperature becomes the most suitable parameter to determine permanence rate of biochar. Research shows that the pyrolysis temperature can influence the life of biochar, with temperatures exceeding 500°C typically resulting in longer half-lives². Here, obtaining direct H/C_{org} measurements is challenging, and adopting data from various sources may lack consistence.

Figure source: Woolf et al., 2021³

Therefore, we calculate permanence rate based on a temperature-based empirical formula developed by Woolf et al³. Here, we have selected the HTT of 550°C, consistent with the data obtained from our pilot project survey. The permanence rate is calculated as 75.08% and assumed the same for biochar produced from all biomass types.

$$per = 0.28 + 8.56 \cdot 0.0001 \cdot T$$

The relevant formulae are marked in red in SI Note 3. Given that this value represents the permanence rate under specific conditions, placing it in the main text might lead to misinterpretation. Therefore, we have removed the specific value and made the revision in line 137-139 of the main text:

‘After pyrolysis, a portion of carbon from the biomass will be sequestered within the biochar and preserved for hundreds of years (Supplementary Note 3). The potential for negative emissions could reach 1.29 GtCO₂ yr⁻¹ under the maximum theoretical scenario.’

References:

1. Budai, A., Zimmerman, A., Cowie, A., Webber, J., Singh, B. P., Glaser, B., Masiello, C., Andersson, D., Shields, F., Lehmann, J., Camps Arbestain, M., Williams, M., Sohi, S., & Joseph, S. (2013). Biochar Carbon Stability Test Method: An assessment of methods to determine biochar carbon stability.
2. Ippolito, J. A., Cui, L., Kammann, C., Wrage-Mönnig, N., Estavillo, J. M., Fuertes-Mendizabal, T., Cayuela, M. L., Sigua, G., Novak, J., Spokas, K., & Borchard, N. (2020). Feedstock choice, pyrolysis temperature and type influence biochar characteristics: A comprehensive meta-data analysis review. *Biochar*, 2(4), 421–438. <https://doi.org/10.1007/s42773-020-00067-x>
3. Woolf, D., Lehmann, J., Ogle, S., Kishimoto-Mo, A. W., McConkey, B., & Baldock, J. (2021). Greenhouse Gas Inventory Model for Biochar Additions to Soil. *Environmental Science & Technology*, 55(21), 14795–14805. <https://doi.org/10.1021/acs.est.1c02425>

2.The period of biochar application will also affect the negative carbon potential of biochar. Is biochar specified in the paper to be applied at a specific time period?

Response:

We greatly appreciate your insightful question, which indeed is of significant interest to us in the next step. As mentioned above, what we have calculated is the upper limit of the supply-side potential. In this context, ‘potential’ means the negative emissions that could be provided when all available biomass feedstocks were used for biochar production. This study focuses on the supply side information, but not on the demand

side or dynamic deployment pathways. Before exploring deployment pathways for biochar in China within an integrated assessment framework, it is essential to investigate the upper limit of its negative emission potential and economic viability at regional level. We aim for this study to also assist policymakers and modelers in gaining a better understanding of the significance of biochar in the context of negative emissions beyond the agronomy value. We hope that the economic evaluation and country-level and subregion-level biochar supply curve we provide can serve as a valuable reference for IAMs considering biochar technology in China.

The development of dynamic deployment pathways is not the primary focus of this paper but rather a subject for future work. This would require in-depth research conducted by multidimensional models (i.e., IAMs) that take into account climate policies, socioeconomic scenarios, technological competition, and other relevant variables. Many factors can influence the specific time period for biochar application, consequently affecting the **actual negative emissions** provided by biochar, distinct from the concept of **negative emission potential**. For example, 1) different time periods may witness varying levels of emission constraints and mitigation difficulty, resulting in varying demands for negative emission technologies (NETs) such as biochar; 2) various NETs compete with each other over time. Currently, there are IAMs that are beginning to explore biomass-based NETs. Many models have examined the timing and scale of deploying another NET - BECCS ¹. Biochar has also gained attention recently and its role is preliminarily explored within IAMs².

In general, what we calculated is the upper limits of negative emission potential of biochar in one year and we did not determine the specific timing for biochar application. However, as you recommended in Comment 3, Shared Socioeconomic Pathways (SSPs) and Representative Concentration Pathways (RCPs) can influence the negative emission potential of biochar by changing the upper limits of available biomass feedstocks. We greatly appreciate your valuable suggestions and have made efforts to enhance our work accordingly, as detailed in response to comment 3.

References:

1. Muratori, M., Bauer, N., Rose, S. K., Wise, M., Daioglou, V., Cui, Y., Kato, E., Gidden, M., Streler, J., Fujimori, S., Sands, R. D., van Vuuren, D. P., & Weyant, J. (2020). EMF-33 insights on bioenergy with carbon capture and storage (BECCS). *Climatic Change*, 163(3), 1621–1637. <https://doi.org/10.1007/s10584-020-02784-5>
2. Fuhrman, J., Bergero, C., Weber, M., Monteith, S., Wang, F. M., Clarens, A. F., Doney, S. C., Shobe, W., & McJeon, H. (2023). Diverse carbon dioxide removal approaches could reduce impacts on the energy–water–land system. *Nature Climate Change*, 13(4), Article 4. <https://doi.org/10.1038/s41558-023-01604-9>

3. In estimating the long-term impact of climate change on the negative carbon potential of biochar, whether the scenarios include a combination of Shared Socioeconomic Pathways (SSPs) and Representative Concentration Pathways (RCPs).

Response:

Thank you very much for your valuable suggestions, which have provided us with clear guidance. In the first round of revisions, we only considered the impact of climate, primarily under RCP scenarios, on the crop yield or forest biomass volume according to existing literature. Given that both socio-economic scenarios and climate scenarios can influence the upper limits of annual availability of biomass resources, and thereby the upper limits of biochar negative emission potential, we consulted the AR6 Scenario Explorer and Database¹. The figure below illustrates the percentage change (vertical axis) in Crop Production and Forestry Production relative to the year 2020 across different scenarios (horizontal axis). Accordingly, under various socio-economic scenarios and climate scenarios, the variation in productions in *Counties of centrally-planned Asia; primarily China* varied between -21% and 76% for crops and between -23% and -6% for forestry. Available biomass resources (i.e., crop residues) can be simplified to be proportional to crop production, thus sharing the same range of variation.

Data source: AR6 Scenario Explorer and Database¹

These scenarios are derived from IAMs, and it is noteworthy that the production estimates for different scenarios exhibit significant variations when accounting for factors such as demand, technological competition, land competition, and trade. Furthermore, some crucial agricultural and land models like GLOBIOM and MAgPIE have not provided data for China in the year 2060 here. According to the latest research,

there is a likelihood of an increase in China's future crop production², rather than a decrease. Given that the primary focus of our paper is not to address demand-side issues, we have refrained from incorporating the above specific value into our calculation. Nevertheless, we think this matter warrants further discussion. As a result, we have implemented the following modifications in SI Note 7:

‘When conducting uncertainty analysis, we considered the influence of climate on biomass resources. However, it should be noted that socio-economic scenarios will also have an impact on biomass production. According to AR6 Scenario Explorer and Database, IAMs that consider both socio-economic scenarios and climate scenarios show that the change of crop production compared to 2020 could range from -21% to 75.5% in 2060 (AIM/CGE, C3IAM, IMAGE), and the change of forestry production could range from -23% to -6 % in 2060 (AIM/CGE, IMAGE), in terms of ‘Counties of centrally-planned Asia; primarily China’. In general, the estimates of production are influenced by complex mechanisms, such as technological competition, carbon pricing, land-use policies, etc., which need to be explored in a broader framework.’

We also emphasized the corresponding text in lines 159-160:

‘Socioeconomic factors also influence future crop production and, consequently, the availability of crop residues, showing significant variations (Supplementary Note 7).’

References:

1. Edward Byers, Volker Krey, Elmar Kriegler, Keywan Riahi, et al. AR6 Scenarios Database hosted by IIASA. International Institute for Applied Systems Analysis, 2022. doi: 10.5281/zenodo.5886911 | url: data.ece.iiasa.ac.at/ar6/

2. Zhao, H., Chang, J., Havlík, P., van Dijk, M., Valin, H., Janssens, C., Ma, L., Bai, Z., Herrero, M., Smith, P., & Obersteiner, M. (2021). China’s future food demand and its implications for trade and environment. *Nature Sustainability*, 4(12), Article 12. <https://doi.org/10.1038/s41893-021-00784-6>

4. The pyrolysis technology and energy consumption used in biochar production have an impact on cost-effectiveness and negative carbon potential. What pyrolysis technology is used? How much is the energy consumption?

Response:

Thank you very much for your questions and suggestions. Here, we will first address the technical aspects and energy consumption, and then discuss the calculations related to syngas and the associated modifications in next response.

In general, we have adopted *the technology of biomass gasification for syngas and biochar co-production*, and taken into account the additional energy consumption.

To be specific, existing biochar production technologies encompass various methods, such as carbonization and co-production techniques. These techniques exhibit significant differences in terms of energy consumption, biochar yield, and other factors, all of which have implications for economic assessments and potential evaluations, as you rightly pointed out.

To ensure that the identified technology type and the benefits from by-products align closely with reality in China, we both conducted literature research and obtained first-hand data from biochar production facilities, which aimed at facilitating a more accurate and reasonable economic evaluation.

Firstly, the technology of biomass gasification for syngas and biochar co-production was selected in this estimation. Despite having a relatively lower weight conversion rate of biochar compared to other technologies, it has demonstrated significant economic benefits from both biochar and syngas, and is considered one of the key industrialization directions for biochar production in China. Some domestic research teams, including Nanjing Agricultural University, Chinese Academy of Agricultural Engineering (Ministry of Agriculture), and Huazhong University of Science and Technology, have been actively involved in this field and performed a series of experiments, as shown in Table S8.

Secondly, our survey on biochar pilot projects also supports the feasibility and development potential of biomass gasification for co-production technology, as shown in Table S9. According to the practical data, we identified the types of by-products and their benefits. Given that in biomass gasification technology, gaseous tar and other products were almost combusted to powering the process, by-product benefits solely come from remaining usable syngas. This system requires a few additional energy consumptions to sustain the operation of the motors. Based on our survey on pilot projects, we have adopted the value of 90 kWh per ton of biomass.

As a result, we have made the following modifications in the manuscript:

1) In lines 172-175 of the main text, we emphasize: ‘Biochar production technology, particularly the technology of biomass gasification for biochar and syngas co-production that is promoted in China (Supplementary Note 2), is both commercially mature and economically competitive, as evidenced by our results’.

2) In SI Note 2, we provided detailed explanation of the technology and supplemented the energy balance equation with additional energy input.

3) In terms of economics evaluation, we accounted for the cost of additional energy input as part of the O&M cost. In SI Note 5 and the second part of the results in the main text, we recalculated data accordingly, and highlighted it in red in the text.

4) In SI Note 6, within the mitigation potential section, we deducted emissions resulting from the additional electricity input:

‘Emissions from electricity consumption during operation and maintenance was deducted (90 kWh/t biomass). The emission factor is 0.581 tCO₂/MWh⁶⁸.’

5. How is the yield of syngas in by-products calculated? Syngas contains CH₄. When estimating the negative carbon potential of biochar, is the reduction of CH₄ emissions from the recovery of syngas taken into account?

Response:

We appreciate your inquiry and have made the following modifications to the calculation of syngas:

To outline the process, we initially calculate the yield and low heating value of various types of biochar. Subsequently, we determine energy efficiency and additional energy input based on a combination of literature data and survey data gathered from pilot projects. Finally, utilizing the parameters of biomass low heating value, biochar low heating value, and energy efficiency, we calculate the heating value of byproducts generated per unit of biomass during the pyrolysis process as follows:

$$Pro_gas_i = (Ade \cdot 0.0036GJ/KWh + Bio_LHV_i) \cdot \mu - Bc_LHV_i \cdot \alpha_i$$

where Pro_gas_i is the production of syngas ($GJ \cdot t^{-1}$ biomass), Ade is additional energy input of pyrolyzing one unit of biomass ($KWh \cdot t^{-1}$ biomass), Bc_LHV_i is the low heating value of biochar ($GJ \cdot t^{-1}$), Bio_LHV_i is the low heating value of biomass ($GJ \cdot t^{-1}$), μ is energy efficiency, and α_i is the weight conversion rate from biomass to biochar.

Detailed information on this calculation can be found in Supplementary Information equation (2-5) to equation (2-9).

In terms of the reduction of CH₄ emissions, we apologize for not providing a clear explanation of our definition of negative emissions. The abstract concept of 'negative emissions' is different from the GHG mitigation potential, the former only consider the removal of carbon dioxide (CO₂) from the atmosphere through technical means¹, of which the carbon sequestration (including storage and its permanence) is one of the crucial features². With reference to Lehmann et al.³, for biochar, negative emissions are 'only delivered in four direct ways: (1) the higher persistence of pyrolysed compared with unpyrolysed biomass; (2) increased growth of plants in soils to which biochar was added, if this increased biomass is itself converted into biochar or other long-lived carbon products; (3) reduced mineralization of the existing soil organic carbon (SOC) together with an increased retention of new plant residue inputs (often called negative priming; and (4) Carbon capture and storage (CCS) of pyrolysis gases and liquids'.

Therefore, it is the permanent fixed C that matters in the concept of 'negative emissions'. It is likely that syngas will be combusted soon for energy use, leading to the conversion of CH₄ into CO₂. CO₂ will be then returned to the atmosphere. This process does not realize C sequestration. Indeed, CCS mentioned above can be applied to syngas, and C

can be stored in basins in this way. However, CCS is still in the early stages of development and carries substantial uncertainty. When CCS is integrated into a biochar system, the inherent advantages of the biochar system - competitiveness in terms of economics, simplicity and convenience of technology, and relative maturity - may be diminished. Therefore, CCS is not considered here and C in syngas is not accounted in our calculation of negative emission potential.

We have made the following modifications in the manuscript:

1) The definition of negative emissions has been highlighted in lines 395-397: 'In this study, the negative emission potential of biochar refers to the CO₂ fixed in biomass from the atmosphere through photosynthesis, and then transferred and permanently preserved in biochar.'

2) Besides, in assessing the mitigation potential other than negative emission potential, we have taken into account the energy consumption emissions avoided by using syngas. We have made the following modifications in SI Note 6:

'Here, the mitigation potential of biochar included the carbon permanently preserved in biochar, the avoided soil GHG emissions, fossil fuel emission offsets, other life cycle emissions, and the avoided GHG emissions if residues are left in the field rather than used for biochar.'

'Fossil fuel emission offsets refer to the emissions of fossil fuels avoided by using syngas. The emission factor for gasoline is 0.0675 tCO₂/GJ⁴¹. Emissions from electricity consumption during operation and maintenance was deducted (90KWh/t biomass). The emission factor is 0.581 tCO₂/MWh⁶⁸.'

'Finally, the mitigation potential that varies depending on the optimal biochar application rate was determined as 1378–1412 MtCO₂eq, i.e., 50% and 53% greater than the negative emission potential, respectively.'

References:

1. Anderson, K., & Peters, G. (2016). The trouble with negative emissions. *Science*, 354(6309), 182–183. <https://doi.org/10.1126/science.aah4567>
2. Fuss, S., Lamb, W. F., Callaghan, M. W., Hilaire, J., Creutzig, F., Amann, T., Beringer, T., Garcia, W. de O., Hartmann, J., Khanna, T., Luderer, G., Nemet, G. F., Rogelj, J., Smith, P., Vicente, J. L. V., Wilcox, J., Dominguez, M. del M. Z., & Minx, J. C. (2018). Negative emissions—Part 2: Costs, potentials and side effects. *Environmental Research Letters*, 13(6), 063002. <https://doi.org/10.1088/1748-9326/aabf9f>
3. Lehmann, J., Cowie, A., Masiello, C. A., Kammann, C., Woolf, D., Amonette, J. E., Cayuela, M. L., Camps-Arbestain, M., & Whitman, T. (2021). Biochar in climate change mitigation. *Nature Geoscience*, 14(12), 883–892. <https://doi.org/10.1038/s41561-021-00852-8>

REVIEWER COMMENTS

Reviewer #1 (Remarks to the Author):

In response to the revisions made, the manuscript effectively addresses all of my previous concerns. The additional adjustments made to account for ash in the weight balance have been skillfully executed and thoroughly elucidated. Furthermore, the incorporation of Table 1 into the main body of the text enhances the clarity of the study, thereby elevating its overall quality.

I recommend proceeding with the publication

Reviewer #2 (Remarks to the Author):

The manuscript entitled “Biochar has significant negative emission potential to satisfy China’s carbon neutrality target” evaluated the negative emission potential of biochar produced from multiple feedstocks and identified the most cost-effective biomass types and deployment locations in China. It is a complete study, focusing on technological, environmental and economic issues, which could provide guidance for systematic deployment of biochar in China. I suggest this manuscript can be published.

Reviewer #3 (Remarks to the Author):

there are many areas that require attention:

1. Could you further detail the novel elements of your work in the introduction section?
2. The research method section should include a comprehensive overview of the methodology used. Additionally, a graphical illustration or flowchart would help clarify the process.
3. Calibration and validation of the logical framework or model and evaluation the results are important. Also the importance of work must be discussed.
4. The literature review is somehow weak (Improvements needed). By reading through, it is difficult to grasp the key justification for the need of this research.
5. The manuscript needs to clearly elaborate more and show what are the problems with the existing works in the field either globally, regionally, or nationally. Without this, readers would have difficulties in seeing the merit of this paper.

6. Toward the end of the introduction section, the author (s) are requested to show what is missing in these previous works to grasp the real limitations of these studies leading to the motivation of conducting this specific research. Otherwise, it is questionable/dubious when the novelty is considered, authors must underline and stress on the novelty of the paper.

7. Using logical frame work to evaluate the relation between environmental index and biochar planning and management is very important. The validation of the model results is important. Do you validate your result?

8. The methodology for selecting indexes and quantifying them and evaluation in a logical frame work is important. Statistical analysis for validation must describe.

9. Describe the dependent and independent variables. Are there any other variables impacts on water value?

10. The designed flowcharts are not really clear to describe the steps of the model in the manuscript.

11. While the result section is well written, there is limited discussion about this study. This makes the whole part of the discussion weak and poor. Comparing the results to the previous studies is not just enough but also should consider providing the implication of your findings. The author (s) are requested to dig deep into the recent literature (Consulting recent publications) on the topic to discuss the overall results of the study.

12. Although the authors have succinctly summarized the major findings but toward the end, the significance of the research findings was not provided. The major weakness of this section is that there is a lack of concluding remarks based on your findings.

13. Recommendations for future direction/orientation for further research based on the remaining gaps is highly encouraged.

14. Please ensure your conclusions section underscores the scientific value added to your paper and the applicability of your findings/results, as indicated previously. Please revise your conclusion part into more detail. You should enhance your contributions and limitations, underscore the scientific value added to your paper, and/or the applicability of your findings/results and future study in this session.

15. It is suggested to compare the results of the present research with similar studies done before.

16. The paper lacks a punch, and conclusions do not provide significant contributions.

Point-by-point Response to Review Comments on Biochar

Reviewer #1 (Remarks to the Author):

In response to the revisions made, the manuscript effectively addresses all of my previous concerns. The additional adjustments made to account for ash in the weight balance have been skillfully executed and thoroughly elucidated. Furthermore, the incorporation of Table 1 into the main body of the text enhances the clarity of the study, thereby elevating its overall quality. I recommend proceeding with the publication

Response:

Thank you very much for your appreciations and previous valuable comments, which are much helpful in improving our work.

Reviewer #2 (Remarks to the Author):

The manuscript entitled “Biochar has significant negative emission potential to satisfy China’s carbon neutrality target” evaluated the negative emission potential of biochar produced from multiple feedstocks and identified the most cost-effective biomass types and deployment locations in China. It is a complete study, focusing on technological, environmental and economic issues, which could provide guidance for systematic deployment of biochar in China. I suggest this manuscript can be published.

Response:

Thank you very much for your appreciations and previous valuable comments, which are much helpful in improving our work.

Reviewer #3 (Remarks to the Author):

there are many areas that require attention:

1. Could you further detail the novel elements of your work in the introduction section?

Response:

Thank you for your feedback. We appreciate the opportunity to clarify the novel elements of our work, which we believe were already intrinsic to our original manuscript. We have refined our introduction to make these elements more explicit.

Our manuscript indeed presents a unique integration of biochar application insights for negative emission strategies, with a specific focus on the diverse context of China - a perspective not extensively covered in existing literature. In our original submission, we articulated three innovative scenarios that outline the technical, sustainable, and maximal theoretical applications of biochar derived from a comprehensive range of biomass feedstocks, including those frequently omitted in prior studies.

Moreover, our work pioneers in providing a sophisticated spatial analysis that identifies the heterogeneity in biomass types and their geographic distribution. This is a crucial advancement, as it addresses the common underestimation of biochar potential due to oversimplified analyses. The findings from our research offer a nuanced and actionable guide for policymakers regarding biochar deployment, addressing both the supply-side intricacies and the varied agronomic and environmental implications across regions.

We have revised our introduction, specifically from Line 58 to 78, to accentuate these points more explicitly. This amendment not only reinforces the originality of our study but also ensures that the foundational elements of our research are clearly presented.

‘A comprehensive spatial analysis integrating the latest knowledge on biochar's role in negative emissions is in need to provide actionable insights for its deployment in the pursuit of carbon neutrality. On one hand, existing experiments tends to narrowly focus on the properties of specific biochar types or their comparative analysis^{28,29} without a granular estimation of their potential for negative emissions and economic impact. On the other hand, regional studies often fail to account for the diversity of biomass resources and biochar properties, leading to a flawed foundation for deployment strategies.

In China, estimates on biochar have predominantly focused on agricultural residues³⁰⁻³², neglecting significant contributions from forestry and grassland residues, as well as potential energy crops. This oversight results in a chronic underestimation of the country's total biochar potential. Furthermore, variations in the physicochemical properties of biochar derived from different biomass resources^{33,34} are typically overlooked, leading to inaccuracies in economic and emissions assessments. The heterogeneity of spatial factors, such as soil texture, is also commonly disregarded, resulting in biased crop yield benefit estimates^{35,36} and flawed economic evaluations. Recent advancements in data availability from field experiments³⁷, assessment methodologies³⁸, and spatial data resolution³⁹ now permit the inclusion of biomass and spatial heterogeneity in assessments, facilitating a detailed and location-specific evaluation of biochar's negative emissions potential and economic implications. Our study leverages the latest scientific progress to present a novel spatially explicit analysis of biochar potential in China. This analysis recognizes the diversity in biomass types and spatial distribution, addressing the prevalent underestimation of biochar's potential and providing a detailed, actionable framework for policymakers to guide biochar deployment.’

2. The research method section should include a comprehensive overview of the methodology used. Additionally, a graphical illustration or flowchart would help clarify the process.

Response:

We are grateful for your valuable insights regarding the presentation of our research methodology. Upon re-examination of our manuscript, we would like to gently highlight that the details of our methods were indeed thoroughly outlined in our initial

submission. Specifically, the step-by-step procedures were detailed from lines 324 to 337 in the main text, providing a comprehensive overview of the methodology employed.

Nevertheless, understanding the importance of clarity and ease of interpretation, we have taken your suggestion into consideration and have now supplemented our description with a visual aid. To this end, "Figure S10: Framework for Biochar Potential and Economics Evaluation" has been added to Supplementary Note 2, offering a concise yet detailed graphical representation of the methodology flow.

We trust this enhancement will make the methodological underpinnings of our study more accessible and further elucidate the robust framework within which our analysis was conducted.

Figure S10 Framework for Biochar Potential and Economics Evaluation

3. Calibration and validation of the logical framework or model and evaluation the results are important. Also, the importance of work must be discussed.

Response:

Thank you for your attention to the robustness and applicability of our research framework or model. We appreciate your comments and understand the necessity for a comprehensive response to concerns raised in Comments 3, 7, 8, and 9. We assure you that our initial submission did address these aspects; however, we welcome the opportunity to clarify and emphasize the strengths and validation of our methodology.

Our methodology has been designed to be both rigorous and adaptable, capable of integrating the variability inherent in biomass resources as well as the regional

specificities that impact biochar application and its potential benefits. The calibration and validation processes of our logical framework are rooted in current best practices and have been meticulously outlined in the manuscript.

To further clarify these points, we have elaborated on the methodology's calibration, validation, and the overarching importance of our work in the revised manuscript. We have also made sure to cross-reference this information in the relevant sections of the text to ensure that it is easily accessible to readers and reviewers.

As mentioned earlier, the primary focus of our work is on the country-level negative emission potential and economics assessments of biochar while considering the heterogeneity of biochar and spatial factors. To achieve this goal, we need to obtain 1) spatial data of various types of biomass resources and 2) parameters for biochar made from various biomass types, and 3) to ensure parameter comparability. We conducted calculations to acquire spatial-scale data for 16 types of agricultural residues, 10 types of forestry residues, grassland residues, and energy crops to address item 1).

Simultaneously meeting items 2) and 3) is challenging, implying that we need an empirical model, rather than data from single experiment or different experiments that are not comparable. Encouragingly, in 2021, Woolf et al.¹ developed a ‘simple and robust empirical model of the avoided greenhouse gas (GHG) emissions due to addition of biochar to mineral soils’ in the paper entitled ‘*Greenhouse Gas Inventory Model for Biochar Additions to Soil*’. Their method is ‘grounded in a comprehensive analysis of current empirical data, making it a robust method that can be used for many applications including national inventories...’. This framework/model is very suitable for our country-level negative emission estimation, making it possible to realize our goal mentioned above.

We place particular emphasis on the parameters related to negative emissions within this experiential framework, since our study focus on negative emission potential of biochar. To be specific, we calculated carbon content (β_i^{DAF}) of dry ash-free (DAF) biochar through Equation (2-2), and calculated the weight conversion rate (α_i^{DAF}) from biomass to DAF biochar through Equation (2-5), allowing us to further calculate the negative emission potential of biochar.

$$\beta_i^{DAF} = 0.93 - 0.92 * e^{-0.0042*T} \quad (2-2)$$

$$\alpha_i^{DAF} = 0.1261 + 0.5391 * e^{-0.004*T} + 0.2733 * L_i \quad (2-5)$$

These two equations are robust. According to Woolf et al.¹, Equation 2-2 is an exponential regression function of pyrolysis temperature (T in °C) from a meta-analysis with 128 measurements from 26 papers ($R^2 = 0.65$); Equation 2-5 is a function of feedstock lignin content (L) and pyrolysis temperature (T) from another meta-analysis (n = 146 from 18 articles, $R^2 = 0.65$). Detailed information of these two functions has been shown in response to Comment 7&8.

This framework can be applied in our study by using ‘simple parameterizations and readily accessible activity data’ as input - feedstock lignin content (L_i) and pyrolysis temperature (T).

Then, we employed the inputs to better align with the Chinese context. We set the medium temperature 550°C as T according to the pilot project survey and existing literature. Also, the data of L_i in term of the residues of main crops in China - rice husk, rice straw, wheat, maize straw, maize cob - are adopted from chemical composition analysis studies, which either research on typical Chinese varieties or averaged over multiple crop composition results. L_i of the grass residues and forestry residues are adopted from Woolf et al. ¹, and those of miscanthus and sweet sorghum are adopted from Brosse et al. ² and Lu et al. ³. In the subsequent economic calculations, we also consider the ash content. This is primarily because in practical biochar deployment, the resultant biochar (dry ash-free plus ash content) is commonly utilized. All of these can be seen in Supplementary Note 2. Subsequently, utilizing Chinese grid-scale biomass resource data (Supplementary Note 1), we conducted calculations for the negative emission potential of biochar (Supplementary Note 3).

Furthermore, to determine the high heating value (HHV_i) of biochar, we adopted empirical correlations based on proximate analysis that are developed by Qian et al. ⁴ ($R^2 = 0.982$). They selected ten kinds representative biomass to prepare biochar materials, and the correlation has been verified by experimental data and has generalizability. The formula is as follows, where Bc_ash_i is the ash content.

$$HHV_i = -30.3\beta_i^2 + 65.2Bc_ash_i^2 + 55.4\beta_i - 48.5Bc_ash_i + 9.591 \quad (2-6)$$

To ensure that the identified technology type and the benefits from by-products align closely with reality in China, we both conducted literature research and obtained first-hand data from biochar production facilities, which aimed at facilitating a more accurate and reasonable economic evaluation. Syngas production was calculated by following energy balance and was subsequently used for industrial steam generation. More in-depth information can be seen in Supplementary Note 2.

In general, since our objective is to investigate the country-level negative emission potential and economics assessments of biochar while considering the heterogeneity of biochar, it is crucial to compare biochar within a unified framework and then conduct a region-specific potential assessment. The integration of Chinese data with a robust empirical framework enables us to realize our research goal.

We emphasized the significance of our work in main text Line 56-58 and 75-78:

‘Global evaluations of biochar's potential have underscored its critical function as a negative emission technology^{18,26,27}. However, to fully harness its benefits, it is crucial to evaluate the negative emissions potential and economic viability of biochar at regional levels.’

‘Our study leverages the latest scientific progress to present a novel spatial analysis of biochar potential in China. This analysis recognizes the diversity in biomass types and

geographic distribution, addressing the prevalent underestimation of biochar's potential and providing a detailed, actionable framework for policymakers to guide biochar deployment.'

References:

1. Woolf, D., Lehmann, J., Ogle, S., Kishimoto-Mo, A. W., McConkey, B., & Baldock, J. (2021). Greenhouse Gas Inventory Model for Biochar Additions to Soil. *Environmental Science & Technology*, 55(21), 14795–14805. <https://doi.org/10.1021/acs.est.1c02425>
2. Brosse, N., Dufour, A., Meng, X., Sun, Q., & Ragauskas, A. (2012). Miscanthus: A fast-growing crop for biofuels and chemicals production. *Biofuels, Bioproducts and Biorefining*, 6(5), 580–598. <https://doi.org/10.1002/bbb.1353>
3. Lu, W., Zhang, Q., Zhou, H., Liu, C., & Cai, H. (2021). Study on thermal stabilities and mechanical properties of sweet sorghum slag/high density polyethylene composites. *Renewable Energy Resources (in Chinese)*, 39(6), 717–723.
4. Qian, C., Li, Q., Zhang, Z., Wang, X., Hu, J., & Cao, W. (2020). Prediction of higher heating values of biochar from proximate and ultimate analysis. *Fuel*, 265, 116925. <https://doi.org/10.1016/j.fuel.2019.116925>

4. The literature review is somehow weak (Improvements needed). By reading through, it is difficult to grasp the key justification for the need of this research.

Response:

We thank you for your constructive critique regarding the literature review section of our paper. The intention of our original submission was to provide a thorough review, with sections strategically placed in both the introduction and Supplementary Information for comprehensive coverage.

Upon reviewing your feedback, we understand the need for greater clarity in illustrating the necessity of our research within the existing body of work. To this end, we have refined the organization of the literature review to enhance its readability and flow. Furthermore, we have integrated more relevant studies from the Supplementary Information into the main text to underscore the research gap our study addresses.

Specifically, we have enriched our review with the latest and most significant contributions to the field of biochar potential assessment, highlighting works that encompass both global perspectives and those particular to the context of China. These updates ensure that the justification for our research is both explicit and compelling, establishing the critical need for the advancements our study proposes.

The revised and strengthened literature review now features prominently in lines 56 to 79 of the main text, offering a direct and insightful lead-in to the contributions of our research.

1. Woolf, D., Amonette, J. E., Street-Perrott, F. A., Lehmann, J. & Joseph, S. Sustainable biochar to mitigate global climate change. *Nat Commun* 1, 56 (2010). <https://doi.org/10.1038/ncomms1053>.
2. Roe, S. et al. Contribution of the land sector to a 1.5 °C world. *Nat. Clim. Chang.* 9, 817–828 (2019). <https://doi.org/10.1038/s41558-019-0591-9>.
3. Werner, C., Lucht, W., Gerten, D. & Kammann, C. Potential of Land-Neutral Negative Emissions Through Biochar Sequestration. *Earth's Future* 10, e2021EF002583 (2022). <https://doi.org/10.1029/2021EF002583>.
4. Xia, L. et al. Integrated biochar solutions can achieve carbon-neutral staple crop production. *Nat Food* 1–11 (2023). <https://doi.org/10.1038/s43016-023-00694-0>.
5. Yang, Q. et al. Country-level potential of carbon sequestration and environmental benefits by utilizing crop residues for biochar implementation. *Applied Energy* 282, 116275 (2021). <https://doi.org/10.1016/j.apenergy.2020.116275>.
6. Yang, Q. et al. Prospective contributions of biomass pyrolysis to China's 2050 carbon reduction and renewable energy goals. *Nat Commun* 12, 1698 (2021). <https://doi.org/10.1038/s41467-021-21868-z>.

In addition to the above literature related to biochar potential, we also reviewed a series of literature related to biochar characterization, spatial heterogeneity, etc., and these are listed in the Introduction section and in the Supplementary Information. Given that we conduct analysis on biochar potential at regional level, we provided further elucidation on the existing research related to biochar potential assessment in China, as shown in line 65 to 71 in the main text:

‘In China, estimates on biochar have predominantly focused on agricultural residues^{30–32}, neglecting significant contributions from forestry and grassland residues, as well as potential energy crops. This oversight results in a chronic underestimation of the country's total biochar potential. Furthermore, variations in the physicochemical properties of biochar derived from different biomass resources^{33,34} are typically overlooked, leading to inaccuracies in economic and emissions assessments. The heterogeneity of spatial factors, such as soil texture, is also commonly disregarded, resulting in biased crop yield benefit estimates^{35,36} and flawed economic evaluations.’

5. The manuscript needs to clearly elaborate more and show what are the problems with the existing works in the field either globally, regionally, or nationally. Without this, readers would have difficulties in seeing the merit of this paper.

Response:

Thank you for your suggestions. As mentioned in response to comment 4 above, we have made modifications to the literature review in terms of nationally research.

Moreover, we summarize the key studies globally. In Line 56 to 64, we mentioned that ‘Global evaluations of biochar's potential have underscored its critical function as a

negative emission technology^{18,26,27}.’ In Line 60 to 64, we mentioned that ‘On one hand, existing experiments tends to narrowly focus on the properties of specific biochar types or their comparative analysis^{28,29} without a granular estimation of their potential for negative emissions and economic impact. On the other hand, regional studies often fail to account for the diversity of biomass resources and biochar properties, leading to a flawed foundation for deployment strategies.’

6. Toward the end of the introduction section, the author (s) are requested to show what is missing in these previous works to grasp the real limitations of these studies leading to the motivation of conducting this specific research. Otherwise, it is questionable/dubious when the novelty is considered, authors must underline and stress on the novelty of the paper.

Response:

Thank you for your suggestions. In addition to addressing the existing research gaps, we have emphasized the significance of our study in lines 57-60:’ However, to fully harness its benefits, it is crucial to evaluate the negative emissions potential and economic viability of biochar at regional levels. A comprehensive spatial analysis integrating the latest knowledge on biochar's role in negative emissions is in need to provide actionable insights for its deployment in the pursuit of carbon neutrality.’

We also outlined the reasons why our research can address these gaps in lines 71-78: ‘Recent advancements in data availability from field experiments³⁷, assessment methodologies³⁸, and spatial data resolution³⁹ now permit the inclusion of biomass and spatial heterogeneity in assessments, facilitating a detailed and location-specific evaluation of biochar's negative emissions potential and economic implications. Our study leverages the latest scientific progress to present a novel spatially explicit analysis of biochar potential in China. This analysis recognizes the diversity in biomass types and spatial distribution, addressing the prevalent underestimation of biochar's potential and providing a detailed, actionable framework for policymakers to guide biochar deployment.’

7. The methodology for selecting indexes and quantifying them and evaluation in a logical frame work is important. Statistical analysis for validation must describe.

Response:

We greatly appreciate your focus on the methodology of index selection, quantification, and the logical framework's evaluation in our study. We would like to clarify that our original manuscript did indeed contain a detailed description of the methodologies employed. These were presented across various sections in the Supplementary Notes—specifically, Supplementary Note 1 for biomass resource data, Supplementary Note 2 for the explication of key parameters, and Supplementary Note 3 for the calculation methodology concerning the negative emission potential of biochar.

In light of your comments, we have revisited these sections and implemented revisions aimed at enhancing their clarity and explicitness. We believe that the transparency and meticulousness of our approach form the cornerstone of our study's reliability and validity.

Additionally, we have further elucidated the statistical rigor underpinning our methodology, aligning it with the most recent advancements in the field. The sections in Supplementary Information now offer a more in-depth explanation of the statistical analyses applied to three critical aspects: the carbon content determination, the conversion rate assessments, and the heating value calculations.

First, to get the carbon content, we used equation 2-2. According to Woolf et al.¹, Equation 2-2 is an exponential regression function of pyrolysis temperature (T in °C) from a meta-analysis with 128 measurements from 26 papers ($R^2 = 0.65$). This meta-analysis is conducted by Neves et al.² in the paper entitled '*Characterization and prediction of biomass pyrolysis products.*' Neves mentioned that 'The elemental composition of char varies roughly from the one of parent fuel to the one of graphite (i.e. 100% carbon), being highly dependent on the pyrolysis conditions. From the present set of literature data, the carbon content of char increases rapidly with temperature increase'. Equation and figure are shown below, with more details being found in the study of Neves et al.²

$$\beta_i^{DAF} = 0.93 - 0.92 * e^{-0.0042*T} \quad (2-2)$$

Source: Neves et al., 2011

Second, to get weight conversion rate, we used equation 2-5. According to Woolf et al.¹, Equation 2-5 is a function of feedstock lignin content (L) and pyrolysis temperature (T) from another meta-analysis (n = 146 from 18 articles, $R^2 = 0.65$).

$$\alpha_i^{DAF} = 0.1261 + 0.5391 * e^{-0.004*T} + 0.2733 * L_i \quad (2-5)$$

Woolf et al.³, mentioned that 'A preliminary multiple regression was performed on these data using pyrolysis temperature, and feedstock composition (lignin, cellulose, hemicellulose, C, H, O and ash content) as model predictors. Multiple regression showed that temperature and feedstock-lignin content were the most important

predictors of biochar yield, accounting for 50% and 30% of R², respectively... We therefore derived an empirical regression model for biochar yield (Y_{ch}) as a function of pyrolysis temperature T (K) and feedstock lignin mass fraction (L_f) ...' More detailed information can be found in paper entitled '*Biofuels from Pyrolysis in Perspective: Trade-offs between Energy Yields and Soil-Carbon Additions*'.

Source: Woolf et al., 2014

Third, to determine the heating value of biochar, we used equation 2-6 developed by Qian et al.⁴ ($R^2 = 0.982$). They selected ten kinds representative biomass to prepare biochar materials, and the correlation has been verified by experimental data and has generalizability. They mentioned that 'The HHV prediction results were based on proximate analysis data of biochar samples made from ten kinds of biomass, including Ash, VM and FC, and all were based on dry basis. The analytical data used in the independent variable range was $1.21\% \leq \text{Ash} \leq 38.09\%$, $5.26\% \leq \text{VM} \leq 30.28\%$, and $38.03 \leq \text{FC} \leq 92.54\%$. Eq. (10) gave the best prediction result with a R² of 0.982, an AAE of 2.59%, and an ABE of 0.08% in proximate analysis.' The following figure shows *Comparison between predicted and experimental HHV for the developed correlations based on proximate analysis*.

$$HHV_i = -30.3\beta_i^2 + 65.2Bc_{ash_i}^2 + 55.4\beta_i - 48.5Bc_{ash_i} + 9.591(2-6)$$

Source: Qian et al., 2020

Finally, to determine the carbon content, weight conversion rate, and the heating value of biochar produced from various biomass types, we employed the empirical formulas mentioned above, utilizing biomass properties as input. We employed the inputs to better align with the Chinese context. We set the medium temperature 550°C as T according to the pilot project survey and existing literature. The data of input-biomass properties- in term of the residues like rice husk, rice straw, wheat, maize straw, maize cob is adopted from chemical composition analysis studies, which either research on typical Chinese varieties or averaged over multiple crop composition results.

References:

1. Woolf, D., Lehmann, J., Ogle, S., Kishimoto-Mo, A. W., McConkey, B., & Baldock, J. (2021). Greenhouse Gas Inventory Model for Biochar Additions to Soil. *Environmental Science & Technology*, 55(21), 14795–14805. <https://doi.org/10.1021/acs.est.1c02425>
2. Neves, D., Thunman, H., Matos, A., Tarelho, L., & Gómez-Barea, A. (2011). Characterization and prediction of biomass pyrolysis products. *Progress in Energy and Combustion Science*, 37(5), 611–630. <https://doi.org/10.1016/j.pecs.2011.01.001>
3. Woolf, D., Lehmann, J., Fisher, E. M., & Angenent, L. T. (2014). Biofuels from Pyrolysis in Perspective: Trade-offs between Energy Yields and Soil-Carbon Additions. *Environmental Science & Technology*, 48(11), 6492–6499. <https://doi.org/10.1021/es500474q>
4. Qian, C., Li, Q., Zhang, Z., Wang, X., Hu, J., & Cao, W. (2020). Prediction of higher heating values of biochar from proximate and ultimate analysis. *Fuel*, 265, 116925. <https://doi.org/10.1016/j.fuel.2019.116925>
8. Using logical frame work to evaluate the relation between environmental index and

biochar planning and management is very important. The validation of the model results is important. Do you validate your result?

Response:

Thank you for your suggestions. As mentioned in response to Comment 3 and 7, we used the empirical formula developed by Woolf et al.¹, Neves et al.², and Qian et al.³, which have been verified for its robustness, as shown in response to Comment 7.

Please allow us to provide further explanation for the rationale behind using this empirical framework in our study. We employ this empirical framework not to evaluate the relationship between environmental index and biochar planning and management, but to calculate the weight conversion rates, carbon contents, and heating value of biochar produced from various types of biomass feedstocks, with the input being the properties of biomass feedstocks in China.

After obtaining results through the empirical formulas, we also compared these results with existing experimental data, especially those involving different types of biomass feedstocks. This further validates the appropriateness of using this framework in our study. For example, experiments that make comparison of biochar produced from different biomass feedstocks usually find that under same pyrolysis condition, rice-based biochar has similar yields to others in terms of dry-ash-free biochar, but has higher yields than others, mainly due to its significantly higher ash content compared to others⁴⁻⁶. Using the empirical formulas and data of biomass properties, we calculated weight conversion rates from rice straw to dry-ash-free biochar as 22.8% (α_i^{DAF}), which plus the rice-straw ash content of 13.4%, resulting in a total of 36.4%. In other words, the weight conversion rates from rice straw to resultant biochar is 36.4% (α_i). On contrary, α_i^{DAF} is 22.9% for maize cob, while its ash content is merely 3.8%, resulting in α_i of 26.8%. The calculation using the empirical framework harmonizes with experimental findings well. We show the results in SI Note2.

References:

1. Woolf, D., Lehmann, J., Fisher, E. M., & Angenent, L. T. (2014). Biofuels from Pyrolysis in Perspective: Trade-offs between Energy Yields and Soil-Carbon Additions. *Environmental Science & Technology*, 48(11), 6492–6499. <https://doi.org/10.1021/es500474q>
2. Neves, D., Thunman, H., Matos, A., Tarelho, L., & Gómez-Barea, A. (2011). Characterization and prediction of biomass pyrolysis products. *Progress in Energy and Combustion Science*, 37(5), 611–630. <https://doi.org/10.1016/j.pecs.2011.01.001>
3. Qian, C., Li, Q., Zhang, Z., Wang, X., Hu, J., & Cao, W. (2020). Prediction of higher heating values of biochar from proximate and ultimate analysis. *Fuel*, 265, 116925. <https://doi.org/10.1016/j.fuel.2019.116925>

4. Wang, Y., Hu, Y., Zhao, X., Wang, S., & Xing, G. (2013). Comparisons of Biochar Properties from Wood Material and Crop Residues at Different Temperatures and Residence Times. *Energy & Fuels*, 27(10), 5890–5899. <https://doi.org/10.1021/ef400972z>

5. Conz, R. F., Abbruzzini, T. F., Andrade, C. A. de, Milori, D. M. B. P., & Cerri, C. E. P. (2017). Effect of Pyrolysis Temperature and Feedstock Type on Agricultural Properties and Stability of Biochars. *Agricultural Sciences*, 8(9), Article 9. <https://doi.org/10.4236/as.2017.89067>

6. He, X., Liu, Z., Niu, W., Yang, L., Zhou, T., Qin, D., Niu, Z., & Yuan, Q. (2018). Effects of pyrolysis temperature on the physicochemical properties of gas and biochar obtained from pyrolysis of crop residues. *Energy*, 143, 746–756. <https://doi.org/10.1016/j.energy.2017.11.062>

9. Describe the dependent and independent variables. Are there any other variables impacts on water value?

Response:

Thank you for your inquiry. Please allow me to provide further elaboration. As mentioned in response to Comment 3, 7 and 8, the empirical framework we adopted includes regression analysis. In Equation 2-2 developed by Neves et al.¹, the independent variable is temperature, and the dependent variable is carbon content of biochar. In Equation 2-5 developed by Woolf et al.², the independent variable is lignin content and temperature, and the dependent variable is weight conversion rate from biomass to biochar. In Equation 2-6 developed by Qian et al.³, the independent variable is ash content and carbon content, and the dependent variable is high heating value biochar. We have made the detailed explanation in response to Comment 3, 7 and 8.

Furthermore, our paper's primary focus is on assessing the potential of biochar as a negative emission technology in the context of addressing climate change. In addition to CO₂, we also performed calculations that encompass other greenhouse gas emissions such as avoiding fossil fuel emissions and soil greenhouse gas emissions in Supplementary information. Our research does not extend to evaluating the impact of biochar application on other environmental elements beyond greenhouse gas emissions, such as water value.

References:

1. Neves, D., Thunman, H., Matos, A., Tarelho, L., & Gómez-Barea, A. (2011). Characterization and prediction of biomass pyrolysis products. *Progress in Energy and Combustion Science*, 37(5), 611–630. <https://doi.org/10.1016/j.peccs.2011.01.001>

2. Woolf, D., Lehmann, J., Fisher, E. M., & Angenent, L. T. (2014). Biofuels from Pyrolysis in Perspective: Trade-offs between Energy Yields and Soil-Carbon Additions.

3. Qian, C., Li, Q., Zhang, Z., Wang, X., Hu, J., & Cao, W. (2020). Prediction of higher heating values of biochar from proximate and ultimate analysis. *Fuel*, 265, 116925. <https://doi.org/10.1016/j.fuel.2019.116925>

10. The designed flowcharts are not really clear to describe the steps of the model in the manuscript.

Response:

Thank you for your suggestions. We have divided the method framework diagram into three parts, namely 1) scenario setting of available biomass feedstocks, 2) parameter determination of negative emission potential and economic, and 3) spatially explicit analysis to identify the locations most suitable for biochar deployment. The framework diagram has been included in Supplementary Note 2 as follows:

Figure S10 Framework for Biochar Potential and Economics Evaluation

11. While the result section is well written, there is limited discussion about this study. This makes the whole part of the discussion weak and poor. Comparing the results to the previous studies is not just enough but also should consider providing the implication of your findings. The author (s) are requested to dig deep into the recent literature (Consulting recent publications) on the topic to discuss the overall results of the study.

Response:

Thank you for your suggestions. In terms of the potential of biochar prepared from agricultural residues, our results are higher compared to the most recent findings, which is due to the fact that we considered a wider range of crop types. We added the description in lines 299-302: ‘In particular, the negative emission potential and mitigation potential of agricultural residues under the sustainable technical scenario amount to 0.42 Gt CO₂ yr⁻¹ and 0.76 Gt CO₂eq yr⁻¹, exceeding the estimates in existing studies (0.05-0.7 Gt CO₂eq yr⁻¹)³⁰⁻³², mainly due to our consideration of a wider range of crop types.’

In terms of the potential of biochar prepared from all kinds of biomass resources including agricultural residues, grass residues, forestry residues and energy crops, we found that negative emission potential of up to 0.92 Gt CO₂ yr⁻¹ at the net cost of 90 \$ t⁻¹CO₂ in a sustainable manner. We compared this potential with the top-down modeling demands in lines 146 to 152 and we also highlight the implication of this finding: ‘The potential under sustainable technical scenario, meanwhile, could fulfill a negative emission demand of 0.92 GtCO₂ yr⁻¹. Given that the median projection for negative emission demands in China is 0.97 GtCO₂ yr⁻¹ in 2050 or 2060 (see Fig.1c), combined with the carbon sink in managed forests averaging 0.63Gt CO₂ yr⁻¹⁴⁷, biochar stands to play a significant role in achieving negative emissions in accordance with the 1.5 °C target and carbon neutrality, without deploying premature NETs such as BECCS and DACCS.’

We also compared the cost of biochar with other negative emission technologies in lines 171-176: ‘Accordingly, the net cost of negative emissions for biochar from agricultural and forestry residues is capped at <100 \$ t⁻¹CO₂ in China, while the net cost for BECCS is typically 30–400 \$ t⁻¹CO₂¹². Although biochar from energy crops and grass residues is more expensive due to high biomass purchasing cost, they still has an economic advantage over other NETs, such as CO₂ fuels (0–670 \$ t⁻¹CO₂), DACCS (30–1000 \$ t⁻¹CO₂), and microalgae (230–920 \$ t⁻¹CO₂) that might be even more costly^{12,13}.’

Further, existing studies did not comprehensively assess the potential and economics of biochar in China at the grid scale. We highlighted our results in lines 250 to 252: ‘In summary, Central and South China and East China not only are rich in biomass resources but also have lower costs (yellow and orange lines in Fig. 4d), and could be preferentially selected as pilot areas for biochar application.’

12. Although the authors have succinctly summarized the major findings but toward the end, the significance of the research findings was not provided. The major weakness of this section is that there is a lack of concluding remarks based on your findings.

Response:

We appreciate your observations regarding the conclusion of our manuscript and the perceived absence of a definitive statement on the significance of our research findings. We intended our original conclusions to underscore the potential role of biochar in

meeting China's negative emissions targets and its cost advantages compared to other negative emission technologies (NETs).

To address your feedback, we have thoroughly reviewed our concluding section to ensure that the implications of our findings are communicated with the emphasis they deserve:

In the original manuscript, we highlighted biochar's role in complementing China's carbon sink strategies and its cost-effectiveness, as detailed in lines 148-152 and 171-176. Additionally, we identified optimal regions for biochar application based on biomass availability and cost considerations in lines 250-252.

Building upon these statements, we have now elaborated on the overarching significance of these findings in the broader context of global climate goals, especially the near-term NETs potential and recommendation on how to proceed with the biochar pilot.

Line 145-146: Under the current technical scenario, the negative emission potential amounts to 0.43 GtCO₂ yr⁻¹, thereby presenting significant near-term mitigation opportunities.

Line 212-214: Given the high upfront input and uncertain returns, pilot biochar applications could begin with the collection of agricultural and forestry residues with high carbon content and conversion rates.

13. Recommendations for future direction/orientation for further research based on the remaining gaps is highly encouraged.

Response:

We are grateful for your recommendation to articulate the future research directions stemming from our study. We wish to clarify that our initial manuscript did indeed contemplate future research possibilities, as mentioned in lines 320-322: 'Future research could aim to explore these potential trade-offs and co-benefits, offering a more holistic understanding of biochar's role in climate change mitigation^{26,55,56}.'. These lines suggested that upcoming studies could delve into the potential trade-offs and co-benefits of biochar use, thus contributing to a more nuanced understanding of its role in climate change mitigation.

Recognizing the importance of providing clear guidance for future research, we have further refined our discussion section in lines 314-317: 'Lastly, our analysis does not fully address the environmental and socio-economic trade-offs associated with the large-scale deployment of biochar. Increasing biomass demand could potentially result in emissions stemming from land-use changes, intensified competition with food production, and a decline in biodiversity⁵⁴.' The modifications now provide a more explicit commentary on the environmental and socio-economic considerations that warrant careful evaluation in the context of biochar's large-scale implementation. Specifically, we underscore the need for subsequent research to examine the

implications of increased biomass demand, which may have repercussions for land use, food security, and biodiversity.

Our intention is to stimulate a forward-looking research agenda that critically examines these aspects and fosters a comprehensive approach to biochar's application within ecological and societal frameworks.

14. Please ensure your conclusions section underscores the scientific value added to your paper and the applicability of your findings/results, as indicated previously. Please revise your conclusion part into more detail. You should enhance your contributions and limitations, underscore the scientific value added to your paper, and/or the applicability of your findings/results and future study in this session.

Response:

We appreciate your directive to emphasize the scientific contributions and practical relevance of our findings in the conclusions section of our paper. We understand the importance of succinctly articulating the implications of our research and are grateful for the opportunity to enhance the clarity of these aspects.

In our initial submission, we presented key scientific findings, such as the substantial negative emission potential of biochar in China, detailed in lines 261-268. This section of the paper underscores both the quantitative results of our study and the economic viability of biochar production, with specific attention to regional variations that could influence policy decisions.

Additionally, we recognize the significance of guiding future research directions based on our findings. As mentioned in lines 320-322, we have suggested avenues for future research to investigate the complex interplay of trade-offs and co-benefits associated with biochar implementation, which could lead to a more comprehensive understanding of its role in climate change mitigation.

To further address your comments, we have revised the conclusion to more robustly detail our study's contributions and limitations, elucidating the scientific value our work adds to the existing body of knowledge. We have also expanded on the applicability of our results, providing a clearer roadmap for how our findings might inform future studies and policy formulation.

The contributions can be shown in line 259-261 of the main text: 'Our study underscores not only the near-term opportunities but also the sustainable prospects of biochar as an established NET in attaining China's carbon neutrality target.', and in line 265-267: 'Our spatial explicit analysis highlights that region with high potential and low-cost negative emissions largely coincide, primarily in East China and Central and South China.'

For limitations, we have updated the text in line 305-322 of the main text: 'Our study has some limitations. First, our analysis does not encompass all types of biomass resources.....Future research could aim to explore these potential trade-offs and co-

benefits, offering a more holistic understanding of biochar's role in climate change mitigation^{26,55,56}.

To highlight the applicability of the results, we have made changes in line 268-271: 'By offering estimations of the negative emission potential and the economics of biochar, our study can provide recommendations for structured deployment and graded integration of biochar, and provide regional information for the integration of biochar technology into IAMs.'

15. It is suggested to compare the results of the present research with similar studies done before.

Response:

We value your suggestion to benchmark our findings against those of similar studies, and we wish to assure you that such comparisons were indeed present in our original manuscript. For instance, as specified in lines 135-138: 'Consequently, the ensuing area for energy crop cultivation is approximately 50.5 Mhm², yielding 0.66 Gt yr⁻¹. These figures align with existing estimates, which range from 3-185 Mhm²⁴¹⁻⁴³ for area and 0.01 to over 1 Gt yr⁻¹⁴³⁻⁴⁶ for production.' Our estimates for the area available for energy crop cultivation and the potential yield were compared with existing literature, showing consistency with previously reported ranges.

To address your comment comprehensively, we have expanded upon these comparisons within the revised manuscript. In particular, we have elaborated on the comparisons in lines 299-302: 'In particular, the negative emission potential and mitigation potential of agricultural residues under the sustainable technical scenario amount to 0.42 Gt CO₂ yr⁻¹ and 0.76 Gt CO₂eq yr⁻¹, exceeding the estimates in existing studies (0.05-0.7 Gt CO₂eq yr⁻¹)³⁰⁻³², mainly due to our consideration of a wider range of crop types.', where we discuss the negative emission and mitigation potential of agricultural residues under the sustainable technical scenario. Our results, which indicate a higher potential than previously estimated, are attributed to our inclusive assessment of a broader spectrum of crop types.

16. The paper lacks a punch, and conclusions do not provide significant contributions.

Response:

We are thankful for your critique and the opportunity to underscore the significant contributions of our work, which may not have been sufficiently highlighted in our initial conclusions.

Our study is at the forefront of examining biochar as a vehicle for negative emissions, with a special focus on the multifaceted context of China—this particular angle has received limited attention in existing scholarship. The original manuscript details three scenarios that capture a full spectrum of biochar applications. These scenarios

incorporate a wide range of biomass feedstocks, some of which have been overlooked in the past, thus filling a notable gap in the field.

Moreover, our research introduces a comprehensive spatial analysis that accurately depicts the variety of biomass types and their distribution. This approach is instrumental in correcting the common trend of biochar potential being underestimated due to generalized analytical methods. Our findings pave the way for informed policy decisions by providing a thorough and practical outline for the deployment of biochar, which takes into account the intricate supply considerations alongside the diverse agricultural and ecological consequences across different regions.

We have revised our conclusion to better capture the essence and impact of our study, ensuring that the significant contributions are clear and resonate with the importance of the topic. We trust that these amendments will clarify the valuable insights our research brings to the scientific community and to policy formulation.

REVIEWER COMMENTS

Reviewer #3 (Remarks to the Author):

The title must be changed to reflect "to think gobally, to act locally" and to be attractive

16. The paper lacks a punch, and conclusions do not provide significant contributions.

It is unclear on which page the authors response has been incorporated in the text.

17. The author may be benefited from surveying this literature.

<https://doi.org/10.1016/j.jenvman.2023.117429>

Point-by-point Response to Review Comments on Biochar

Reviewer #3 (Remarks to the Author):

1. The title must be changed to reflect "to think globally, to act locally" and to be attractive

Response:

We appreciate your insightful suggestions, which captured the essence of our paper. The previous title of this paper was 'Biochar has significant negative emission potential to satisfy China's carbon neutrality target'. We have revised the title and emphasized 'Geospatial Analysis' to reflect 'locally' in accordance with your recommendation:

Geospatial Analysis Reveals Biochar's Negative Emission Potential for China's Carbon Neutrality Goal

2. The paper lacks a punch, and conclusions do not provide significant contributions. It is unclear on which page the authors response has been incorporated in the text.

Response:

Thank you for your suggestion.

In last revision, we incorporated your review comments in two parts:

In lines 259-263 in last version of revision: 'Our study underscores not only the near-term opportunities but also the sustainable prospects of biochar as an established NET in attaining China's carbon neutrality target. We found that biochar presents considerable negative emission potential within China, with the current technical and sustainable technical negative emission potential being 0.43 and 0.92 Gt CO₂ per annum, respectively.'

In lines 270-273 in last version of revision: 'By offering estimations of the negative emission potential and the economics of biochar, our study can provide recommendations for structured deployment and graded integration of biochar, and provide regional information for the integration of biochar technology into IAMs.'

We further incorporated your review comments in this version as follows:

In lines 265-270, we emphasized 'spatially explicit analysis' in this version and highlighted related conclusion: 'Our high-resolution estimates of biochar's potential yield valuable insights that inform decision-making regarding investments and production in this sector. Our spatially explicit analysis highlights that region with high potential and low-cost negative emissions largely coincide, primarily in East China and Central and South China. Remarkably, a few areas have the potential to achieve positive returns due to high crop yields or crop value.'

We trust that these amendments will clarify the valuable insights our research brings to the scientific community and to policy formulation.

3. The author may be benefited from surveying this literature. <https://doi.org/10.1016/j.jenvman.2023.117429>

Response:

Thank you for your suggestion. We have added a citation to this literature at lines 45-46 in the main text: ‘Biochar has ancient origins in Amazonian farmlands and it has existed for centuries^{19,20}, although it has gained recognition in the field of climate change mitigation in only the previous two decades^{21,22}.’

References:

22. Kurniawan, T. A. et al. Challenges and opportunities for biochar to promote circular economy and carbon neutrality. *Journal of Environmental Management* 332, 117429 (2023). <https://doi.org/10.1016/j.jenvman.2023.117429>

REVIEWERS' COMMENTS

Reviewer #3 (Remarks to the Author):

well done